# From Static Benchmarks to Dynamic Protocol: Agent-Centric Text Anomaly Detection for Evaluating LLM Reasoning

**Seungdong Yoa**[1], **Sanghyu Yoon**[1], **Suhee Yoon**[1], **Dongmin Kim**[1], **Ye Seul Sim**[1],
**Junhyun Lee**[2*], **Woohyung Lim**[1*]
[1]LG AI Research, [2]Division of Computer Engineering, Hankuk University of Foreign Studies
{seungdong.yoa, sanghyu.yoon, suhee.yoon, dmkim}@lgresearch.ai
{ysl.sim, w.lim}@lgresearch.ai, junhyun.lee@hufs.ac.kr

## Abstract

The evaluation of large language models (LLMs) has predominantly relied on static datasets, which offer limited scalability and fail to capture the evolving reasoning capabilities of recent models. To overcome these limitations, we propose an agent-centric benchmarking paradigm that moves beyond static datasets by introducing a dynamic protocol in which autonomous agents iteratively generate, validate, and solve problems. Within this protocol, a teacher agent generates candidate problems, an orchestrator agent rigorously verifies their validity and guards against adversarial attacks, and a student agent attempts to solve the validated problems. An invalid problem is revised by the teacher agent until it passes validation. If the student correctly solves the problem, the orchestrator prompts the teacher to generate more challenging variants. Consequently, the benchmark scales in difficulty automatically as more capable agents are substituted into any role, enabling progressive evaluation of large language models without manually curated datasets. Adopting text anomaly detection as our primary evaluation format, which demands cross-sentence logical inference and resists pattern-matching shortcuts, we demonstrate that this protocol systematically exposes corner-case reasoning errors that conventional benchmarks fail to reveal. We further advocate evaluating systems along several complementary axes including cross-model pairwise performance and progress between the initial and orchestrator-finalized problems. By shifting the focus *from fixed datasets to dynamic protocols*, our approach offers a sustainable direction for evaluating ever-evolving language models and introduces a research agenda centered on the co-evolution of agent-centric benchmarks.

## 1 Introduction

Static benchmarks, such as MMLU Hendrycks et al. (2021), GSM8K Cobbe et al. (2021) and Big-Bench Srivastava et al. (2023), once served as reliable indicators of language model progress. However, frontier large language models (LLMs) now approach—or even surpass—human-level accuracy on many of these tasks Pu et al. (2023); Maslej et al. (2024); Phan et al. (2025). Because these benchmark suites are finite, publicly accessible, and often included in pretraining corpora, models may inadvertently memorize substantial portions of the test data Deng et al. (2024). This can lead to inflated leaderboard results that do not reflect genuine improvements in reasoning ability. Unfortunately, it has become increasingly difficult to draw meaningful distinctions from these overused datasets. First, data contamination is now common: large-scale data collection often includes benchmark questions in pretraining datasets, and efforts to remove them afterward are usually incomplete Raji et al. (2021). Second, because static benchmarks contain a limited number of items, model developers may—sometimes without realizing it—tune their systems to match the details of these benchmarks. This creates feedback loops that improve scores without real gains in general reasoning ability Dodge et al. (2020). Third, once a benchmark is considered "solved", the research community must quickly create a new one. This leads to a cycle of rapid creation and decline,

---

*Corresponding authors

Figure 1: **Comparison of text anomaly samples.** *Left*: Existing benchmarks include obvious anomalies (e.g., complete off-topic from sports news to economy news) that are clear but too trivial. *Right*: ATAD examples introduce subtle shifts within context (e.g., benefits to ethics in healthcare AI), preserving clarity while presenting reasoning-intensive challenges. Our collaborative agents resolve the clarity-difficulty trade-off through iterative task refinement.

which uses up valuable time and provides only short-term insight into model performance Rogers (2021). These limitations highlight the inherent shortcomings of static benchmarks in evaluating real reasoning capabilities.

To overcome these shortcomings, dynamic benchmarks for LLMs are essential as they continuously evolve, mitigating data contamination and preventing models from overfitting to finite test sets. In particular, text anomaly detection serves as a powerful task to reveal subtle reasoning flaws, providing clearer insight into the true capabilities and limitations of LLMs Maimon & Tsarfaty (2023). However, constructing high-quality text anomaly detection problems remains challenging: increasing the difficulty often sacrifices clarity, while ensuring clarity typically results in overly simple tasks. Figure 1 illustrates this trade-off and motivates our protocol's design. We introduce the **A**gent-centric **T**ext **A**nomaly **D**etection (**ATAD**), a benchmark protocol that replaces the static-dataset paradigm with a three-agent system. In this protocol, as illustrated in Figure 2, a teacher agent generates candidate problems, an orchestrator agent validates them and filters out defective items, and a student agent attempts to solve the qualified problems. As a problem format, reasoning-centric anomaly detection tasks are well suited for evaluating LLMs: they require cross-sentence logical inference, resist pattern-matching shortcuts and training data leakage, and support objective, fine-grained scoring. Asking a model to identify and explain the single sentence that disrupts a passage's coherence offers a precise and robust measure of reasoning ability—one that is less prone to exploitation than many existing benchmarks. By shifting the focus *from fixed datasets to dynamic protocols*, we offer a sustainable direction for evaluating ever-evolving language models and invite the community to explore a research agenda in which models and the benchmarks that probe them co-evolve. We will release an open-source reference implementation with empirical results showing that ATAD surfaces reasoning weaknesses invisible to static benchmarks. A comprehensive discussion of related work on dynamic benchmarking and text anomaly detection is provided in Appendix.

## 2 ATAD: BENCHMARK PROTOCOL DESIGN AND OPERATION

We introduce a novel agent-centric dynamic benchmarking protocol, Agent-Centric Text Anomaly Detection (ATAD), illustrated in Figure 2. ATAD is designed to construct an adaptive benchmark for text anomaly detection by leveraging a teacher-student competitive loop and an orchestrator-regulated validation mechanism. Unlike static datasets, our protocol dynamically evolves problem difficulty based on student model performance while ensuring clarity and fairness through rigorous validation. This design enables the benchmark to scale with the capabilities of emerging language models, supporting sustainable and progressively challenging evaluation over time.

### 2.1 AGENT ROLES

**Teacher Agent**: Generates problems and increases their difficulty when the Student solves them correctly, forming a competitive loop that adapts to the Student's capabilities.
**Orchestrator Agent**: Validates the generated problem to ensure it is well-formed, unambiguous, aligned with the expected task type, and free from adversarial design. It also checks whether the problem is logically coherent and appropriately matches the intended difficulty level.
**Student Agent**: Attempts to solve the validated problem. If it succeeds, the problem is made harder; if it fails, the problem is accepted into the benchmark.

The naming of Teacher and Student refers to agent roles in the protocol and is unrelated to model training paradigms such as knowledge distillation. In our framework, the competitive interaction between the Teacher and Student agents is leveraged to drive difficulty escalation in benchmark

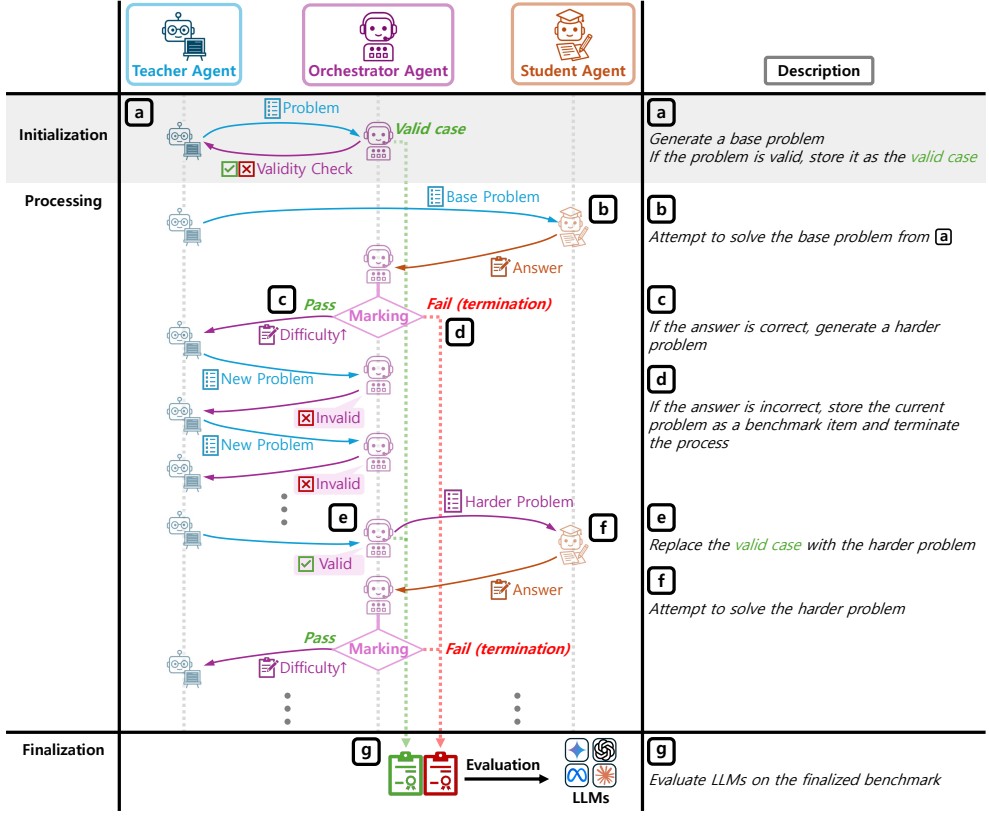

Figure 2: **Illustration of the overall ATAD protocol.** Three agents iteratively interact to generate progressively challenging benchmarks designed to uncover subtle reasoning weaknesses in LLMs.

construction. This dynamic, however, can risk generating ambiguous or adversarial problems in the pursuit of harder samples. To mitigate this, the Orchestrator agent plays a crucial role in ensuring quality and fairness at each iteration. This validation process is particularly important for tasks like text anomaly detection, where subtle shifts in coherence, semantics, or phrasing can easily compromise problem clarity.

## 2.2 PROTOCOL PHASES

Our proposed benchmark construction protocol operates through a multi-agent system involving a Teacher, an Orchestrator, and a Student agent. These agents interact through two core phases: the Initialization Phase and the Adaptive Difficulty Scaling Phase. Each phase features automatic iteration control mediated by the Orchestrator. A visual summary of the protocol workflow is provided in Figure 2, with steps annotated from **a** to **g**.

### 2.2.1 INITIALIZATION PHASE (BASE PROBLEM GENERATION)

The protocol begins with the Teacher agent generating a base-level problem for a designated text anomaly detection task (e.g., semantic deviation, sentence order inconsistency), corresponding to the **label a** in Figure 2. These base problems are intended to be of low difficulty and serve as the starting point for the benchmark construction.

Each generated problem is submitted to the Orchestrator for a multi-criteria validation process. The Orchestrator evaluates the sample for well-formedness, clarity, logical coherence, task type adherence, and fairness, while guarding against adversarial design or unanswerable ambiguity.

If the problem is invalid, the Orchestrator returns detailed feedback to the Teacher, prompting regeneration. This loop is governed by the Orchestrator's validation decisions and continues until a valid problem is produced or a maximum number of attempts (`max_init_loops`) is reached.

Once the problem passes validation, it is stored as a valid base problem and passed on to the Adaptive Difficulty Scaling Phase.

### 2.2.2 ADAPTIVE DIFFICULTY SCALING PHASE

This phase begins with the Student's first attempt at the validated base problem and corresponds to the **b** through **f** labels in Figure 2. The Student attempts to solve the base problem (**label b**). If the Student fails, the problem is finalized as a benchmark item (**label d**), as it exposes a limitation in the Student's current reasoning capacity.

If the Student succeeds, the Orchestrator prompts the Teacher to generate a more challenging variant of the problem (**label c**). The Teacher, informed by the Student's prior success, creates a harder version aimed at pushing the Student's capabilities further. This new problem undergoes the same validation process by the Orchestrator to ensure that difficulty has increased meaningfully without compromising task clarity or fairness (**label e**).

Once validated, the harder problem replaces the previous one and is presented to the Student for another attempt (**label f**). This cycle—solving, regenerating, validating—continues iteratively until the Student fails or the iteration cap (`max_student_loops`) is reached. If the Teacher's harder problem is rejected by the Orchestrator, it may be prompted to slightly reduce the difficulty and regenerate, preserving the same task structure while avoiding ambiguity or excessive complexity. Although this does not constitute a formal decrease in the difficulty level, it allows for iterative refinement within the same hardness tier. If multiple regeneration attempts fail to produce a valid harder problem, the process terminates with the last previously validated problem—typically the one that the Student successfully solved—being finalized as the benchmark item.

The most difficult validated problem that causes the Student to fail is adopted as the finalized benchmark item. This structure allows the benchmark to automatically calibrate difficulty per instance, producing finely tuned evaluation samples based on actual model behavior.

### 2.2.3 EVALUATION PHASE

This phase corresponds to the **label g** in Figure 2. After benchmark samples are finalized through the above process, LLMs can be evaluated using the curated benchmark. Each problem is associated with its final difficulty level and validation metadata, supporting both overall performance comparisons and fine-grained reasoning diagnostics.

### 2.3 KEY FEATURES

Our benchmarking framework is grounded in two complementary principles: a competitive protocol in which the Teacher challenges the Student with progressively harder problems, and an adaptive validation mechanism where the Orchestrator ensures that difficulty scaling remains fair, coherent, and well-formed. Together, these two dynamics enable ATAD to produce reliable, high-quality benchmarks tailored to a model's actual reasoning capacity.

**Difficulty Scaling via Teacher-Student Competition.** The Teacher agent is implicitly incentivized to analyze the Student's prior successes and failures. This allows it to generate novel problems that directly target the Student's weaknesses or extend beyond its current competence, yielding more sophisticated samples than mere perturbations of existing items. Difficulty is adjusted dynamically based on the Student's performance, forming a competitive loop that drives benchmark depth.

**Orchestrator-Regulated Difficulty Control.** To prevent uncontrolled or adversarial difficulty escalation, the Orchestrator agent validates each problem before it is presented to the Student. It checks logical coherence, task adherence, clarity, and difficulty appropriateness, and autonomously decides whether the Teacher should regenerate a sample. This ensures that problem progression remains both challenging and fair, balancing the Teacher's incentives with principled quality control.

**Autonomous Iteration Control.** Unlike benchmarks with fixed iteration schedules, ATAD relies on the Orchestrator to dynamically determine when the Teacher should regenerate a problem or proceed to evaluation. This mechanism replaces manual tuning with agent-driven adaptability, ensuring high-quality, context-appropriate problems at every step.

**Failure-Driven Sample Finalization.** Problems are finalized not at creation, but at the point of Student failure. This empirical approach anchors benchmark difficulty in actual model limitations

rather than manual labels, surfacing failure cases that are often missed in static datasets.

**Dynamic Difficulty Localization.** Unlike benchmarks that assign difficulty globally, ATAD adjusts difficulty at the instance level based on Student feedback. This enables precise, localized probing of reasoning weaknesses and model-specific blind spots.

**Cross-Agent Instantiability.** ATAD is modular by design and supports different model pairings (e.g., $\text{ATAD}^{\text{gpt-4o}}_{\text{gemini2-flash}}$), enabling comparative evaluation and tracking of model evolution over time.

**Broad Task Coverage.** Our benchmark spans seven types of text anomaly detection tasks (see Section 3.2), capturing a wide range of reasoning capabilities including discourse coherence, contradiction detection, referential clarity, and stylistic consistency.

## 3 TASK DESIGN FOR TEXT ANOMALY DETECTION

This section presents our design of text anomaly detection tasks as a probe of LLM reasoning (Section 3.1) and introduces a taxonomy of seven anomaly types (Section 3.2).

### 3.1 TASK OVERVIEW AND MOTIVATION

We identify text anomaly detection as a particularly suitable domain for evaluating the reasoning capabilities of LLMs. These tasks target subtle inconsistencies in logic, coherence, or semantics, requiring genuine cross-sentence inference and resisting shortcuts based on surface-level patterns. However, creating high-quality text anomaly problems remains challenging: increasing task difficulty often introduces ambiguity, while prioritizing clarity can lead to trivial or shallow problems. This trade-off is especially pronounced in language-based tasks, where, unlike math or science, answers lack grounding in formal rules. Yet standardized exams like the GRE, GMAT, and LSAT show that natural language questions can still demand structured reasoning with clear answer standards. Inspired by these formats, our benchmark emphasizes deep reasoning while maintaining clarity and objectivity. Still, generating such problems at scale—especially in text anomaly detection—remains difficult, as it requires balancing subtlety and unambiguity. Our adaptive benchmarking protocol addresses this via a teacher-student competition regulated by an orchestrator, forming a self-calibrating system that reliably surfaces nuanced reasoning failures in LLMs.

### 3.2 TASK TAXONOMY: SEVEN TYPES OF TEXT ANOMALIES AND REASONING SKILLS TARGETED

Each task in our taxonomy is designed to assess a distinct aspect of LLM reasoning, such as coherence, logical consistency, or ambiguity resolution—areas often underrepresented in existing benchmarks. Together, the seven task types provide a broad and fine-grained evaluation of language understanding. While each task targets a core reasoning capability, we further diversify the benchmark by selectively incorporating anomaly factors known to challenge LLMs, including subtle semantic shifts or structural inconsistencies. These additions are applied to a subset of examples to enhance difficulty without sacrificing clarity or task diversity.

**T1. Sentence Context Anomaly** targets *contextual reasoning*, requiring the model to detect semantic inconsistencies between individual sentences and the paragraph's main theme. Challenge factors include *minor topic shifts* and *semantic deviations* that appear grammatically well-formed but subtly disrupt thematic coherence.

**T2. Paragraph Order Consistency** assesses *discourse coherence* by determining the correct order of sentences based on topic flow, causal and temporal dependencies. Challenge factors involve *sentence reordering* that appears locally coherent but requires comprehensive understanding of global document structure to detect.

**T3. Blank-based Choice Anomaly** requires both *lexical* and *pragmatic reasoning* to identify an inappropriate word or phrase within context. Challenge factors focus on *lexical fit* and *collocation*, requiring the detection of choices that are grammatically correct but contextually inappropriate. This demands both common sense and sensitivity to subtle nuances.

**T4. Bridge Sentence Evaluation** focuses on *logical bridging* and *topic shift detection*, requiring the model to judge whether a candidate sentence logically connects two related paragraphs. Challenge factors include *weak logical connections* and *abrupt topic shifts*, where the sentence itself may seem plausible but fails to maintain coherent discourse flow.

**T5. Referential Ambiguity** tests *coreference resolution* to identify sentences where pronouns or referring expressions are ambiguous or misleading, disrupting clarity in discourse interpretation.

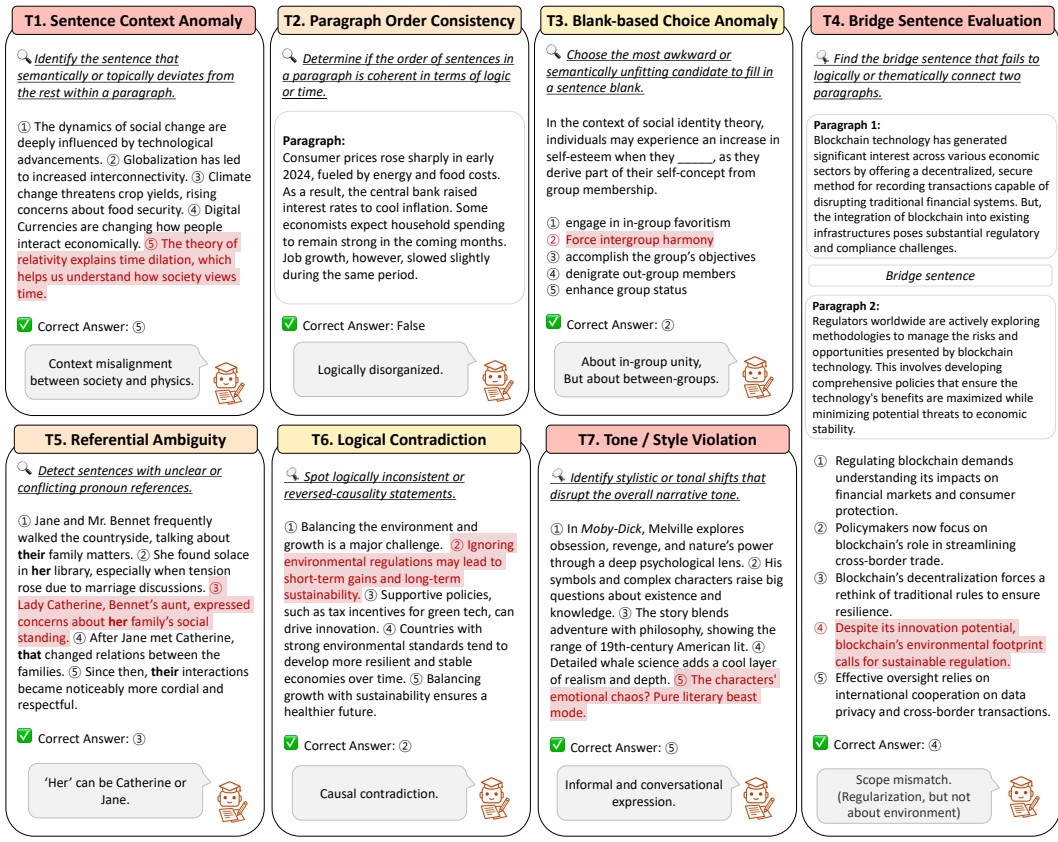

Figure 3: **Examples of the seven task types of text anomalies.** With the exception of T2, each task requires identifying a guaranteed anomaly within the sample (e.g., by selecting a sentence or choice), rather than performing a simple binary classification.

Challenge factors involve *ambiguous pronouns* and *unclear references* that disrupt sentence clarity. **T6. Logical Contradiction** measures *causal and contradiction reasoning*. The model detects inconsistencies such as violated cause–effect relationships or misinterpreted correlations as causation. Challenge factors include *contradictory claims* and *causal reversals*.

**T7. Tone/Style Violation** evaluates *stylistic reasoning* by assessing whether all sentences maintain a consistent tone and register (e.g., formal vs. informal). The model must identify any sentence that deviates from the overall style. Challenge factors include *tone shifts* and *register mismatches* that subtly undermine stylistic coherence.

While each task focuses on a primary reasoning skill, practical cases often demand the integration of multiple capabilities—such as critical thinking and fine-grained semantic analysis. For example, T4 not only requires assessing logical coherence but also detailed semantic understanding. Building on this, we enhance task diversity by incorporating them within six academic domains frequently found in standardized reasoning exams (*e.g.,* GRE, LSAT), including science, philosophy, politics/society, psychology, economics, and literature. Rather than assigning domains randomly, we systematically align them to the tasks where the domain's inherent characteristics amplify reasoning challenges. This principled topic-to-task mapping is detailed in Appendix, with additional examples and domain-specific motivations. Figure 3 outlines representative task formats, with full design details available in Appendix as well.

# 4 EXPERIMENTS AND RESULTS

This section presents our experimental evaluation of the benchmark generated through our protocol, highlighting its utility for assessing LLM reasoning. We evaluate overall performance, examine the Teacher-Student competition protocol for difficulty scaling, and assess the contribution of Orchestrator

Table 1: Overall Performance of LLMs on our Text Anomaly Detection Benchmark. Average accuracy of each LLM, across the four datasets generated by four agent families (GPT, Gemini, Claude, LLaMA), is shown for each anomaly type (T1-T7) and overall.

| Evaluation Model | T1 | T2 | T3 | T4 | T5 | T6 | T7 | Avg. |
|---|---|---|---|---|---|---|---|---|
| GPT-3.5-Turbo OpenAI (2024a) | 59.00 | 16.00 | 66.75 | 48.50 | 55.75 | 51.75 | 81.50 | 54.18 |
| GPT-4o-mini OpenAI (2024c) | 57.25 | 17.00 | 62.50 | 54.00 | 52.50 | 58.75 | 83.00 | 55.00 |
| GPT-4o OpenAI (2024b) | 62.00 | 21.25 | 68.25 | 53.25 | 49.25 | 56.75 | 81.00 | 55.96 |
| GPT-o4-mini OpenAI (2024d) | 63.25 | 30.25 | **68.50** | 53.00 | 47.25 | 57.25 | 80.00 | 57.07 |
| Gemini-1.5-Flash Reid et al. (2024) | 6.00 | 11.25 | 62.00 | 48.75 | 17.50 | 10.75 | 21.00 | 25.32 |
| Gemini-2.0-Flash-Lite Google DeepMind (2024) | 64.00 | 10.75 | 63.50 | 52.25 | **62.75** | **62.00** | 86.25 | 57.36 |
| Gemini-2.0-Flash Google DeepMind (2024) | 65.25 | 25.00 | 63.00 | 58.25 | 51.00 | **62.00** | **88.00** | 58.93 |
| Claude-3-Haiku Anthropic (2024b) | 63.75 | 12.00 | 61.00 | 51.75 | 53.50 | 60.00 | 72.75 | 53.54 |
| Claude-3.5-Haiku Anthropic (2024a) | 19.75 | **55.00** | 7.25 | 5.00 | 5.50 | 8.50 | 35.50 | 19.50 |
| Claude-3.5-Sonnet Anthropic (2024a) | **65.75** | 31.75 | 65.00 | 59.50 | 53.50 | 57.50 | 86.75 | **59.96** |
| LLaMA-3.1-8B Llama Team (2024) | 39.50 | 12.75 | 35.50 | 24.50 | 53.00 | 38.75 | 68.75 | 38.96 |
| LLaMA-3.3-70B Llama Team (2024) | 60.75 | 27.75 | 63.25 | **60.00** | 52.25 | 57.75 | 84.25 | 58.00 |

validation. Additionally, we explore its use in forecasting future LLM capabilities and test its consistency across multiple runs.

## 4.1 EVALUATION SETUP

To evaluate LLM performance on our text anomaly benchmark, we established the following setup:

**Benchmark dataset.** The benchmark dataset comprises 700 samples per generation model, with 100 instances for each of the seven task types.

**Generation models.** We used the following LLMs as Teacher, Student, and Orchestrator agents to generate the benchmark datasets: GPT-4o OpenAI (2024b), Claude-3.5-Sonnet Anthropic (2024a), Gemini-2.0-Flash Google DeepMind (2024), and LLaMA-3.3-70B Llama Team (2024). (When not explicitly stated, the Teacher, Student, and Orchestrator agents within a generation process use the same LLM.)

**Evaluation models.** We evaluated the generated datasets using a diverse set of LLMs: GPT-3.5-turbo, GPT-4o-mini, GPT-4o, GPT-o4-mini, Claude-3.0-Haiku, Claude-3.5-Haiku, Claude-3.5-Sonnet, Gemini-1.5-Flash, Gemini-2.0-Flash-Lite, and Gemini-2.0-Flash. These models serve as our baseline for assessing the difficulty and effectiveness of the benchmark.

## 4.2 OVERALL PERFORMANCE EVALUATION

Table 1 presents the overall performance of various LLMs on our text anomaly detection benchmark. We report accuracy as the primary evaluation metric, calculated as the proportion of correctly identified anomalies. Table 1 showcases the average accuracy achieved by each evaluation model across the four distinct benchmark datasets, each generated by a different agent family: GPT, Gemini, Claude, and LLaMA. For the benchmark generation process, the Teacher, Student, and Orchestrator agents were configured to be the same LLM for simplicity (e.g., GPT-4o for all three roles within the GPT-generated benchmark).

The results reveal a varied landscape of performance across different anomaly types (T1–T7). Notably, no single evaluation model consistently outperformed others across all categories, suggesting that the nature of the anomaly significantly influences detection accuracy. Claude-3.5-Sonnet achieved the highest overall average accuracy (59.96%), indicating strong general capability. However, other models surpassed Claude on specific types: GPT-4o-mini outperformed Claude on T3 by 3.5%, and Gemini-2.0-Flash exceeded Claude on T6 by 4.5%. Interestingly, certain evaluation models showed remarkable proficiency in specific anomaly types. Claude-3.5-Haiku, despite its relatively lower overall average (53.54%), achieved the highest accuracy in detecting anomalies of type T2 (55.00%). This highlights the potential for certain models to possess specialized strengths in identifying particular kinds of textual irregularities. While the overall average accuracy across all models and anomaly types indicates the inherent difficulty of the task, the varying performance across different anomaly types underscores the benchmark's ability to probe diverse aspects of LLM understanding and reasoning regarding text anomalies.

## 4.3 VALID DIFFICULTY SCALING VIA COMPETITIVE AGENTS

To assess whether our competitive protocol effectively scales problem difficulty, we compare evaluation model performance on the initial base problems and the finalized benchmark versions. Table 2 presents the average accuracy of each evaluation model, computed across the seven anomaly types

Table 2: Comparison of the LLMs' performance on the initial (base) datasets, consisting of the base problems, and the final versions of the benchmark datasets. Each column represents a different dataset, generated by GPT-4o, Claude-3.5-Sonnet, Gemini-2.0-Flash, and LLaMA-3.3-70B, respectively. The observed performance drop from base to final problems highlights the effectiveness of ATAD in exposing the weaknesses of LLM reasoning.

| Evaluation Model | GPT-4o | | Gemini-2.0-Flash | | Claude-3.5-Sonnet | | LLaMA-3.3-70B | |
|---|---|---|---|---|---|---|---|---|
| | Base | Final | Base | Final | Base | Final | Base | Final |
| GPT-3.5-turbo | 91.00 | 67.71 | 80.00 | 42.00 | 83.71 | 61.43 | 86.00 | 45.57 |
| GPT-4o-mini | 93.00 | 68.29 | 80.43 | 42.71 | 84.14 | 57.86 | 87.43 | 51.14 |
| GPT-4o | 94.29 | 72.43 | 83.29 | 44.71 | 87.29 | 62.71 | 89.29 | 44.00 |
| GPT-o4-mini | 91.86 | 72.43 | 83.57 | 47.14 | 87.29 | 61.86 | 87.71 | 46.86 |
| Gemini-1.5-Flash | 50.57 | 30.29 | 40.14 | 17.00 | 40.43 | 28.29 | 41.43 | 25.71 |
| Gemini-2.0-Flash-lite | 92.43 | 69.14 | 81.57 | 45.43 | 83.86 | 58.86 | 85.14 | 56.00 |
| Gemini-2.0-Flash | 92.29 | 71.86 | 82.43 | 44.29 | 85.43 | 61.86 | 88.00 | 57.71 |
| Claude-3-Haiku | 91.57 | 67.86 | 79.43 | 42.86 | 82.71 | 54.57 | 83.43 | 48.86 |
| Claude-3.5-Haiku | 36.86 | 18.86 | 39.86 | 24.71 | 39.71 | 18.57 | 45.86 | 15.86 |
| Claude-3.5-Sonnet | 91.71 | 72.86 | 83.86 | 47.43 | 88.86 | 63.29 | 88.29 | 56.29 |
| LLaMA-3.1-8B | 67.29 | 47.00 | 59.57 | 28.57 | 63.57 | 33.57 | 64.14 | 46.71 |
| LLaMA-3.3-70B | 93.43 | 72.43 | 82.71 | 43.57 | 89.29 | 64.57 | 92.43 | 51.43 |

Table 3: Comparison of LLMs' Performance and Problem Quality on the benchmark generated by GPT-4o agents. Problem quality is evaluated by each model acting as a reviewer, comparing benchmarks generated with and without the use of an Orchestrator.

| Evaluation Model | Performance (%) | | Problem Quality | | | | | | | |
|---|---|---|---|---|---|---|---|---|---|---|
| | w/o Orch. | w/ Orch. | Validity (1–5) | | Coherence (1–5) | | Fairness (1–5) | | Approval Rate (%) | |
| | | | w/o Orch. | w/ Orch. | w/o Orch. | w/ Orch. | w/o Orch. | w/ Orch. | w/o Orch. | w/ Orch. |
| GPT-4o | 68.29 | 72.43 | 4.30 | 4.85 | 3.71 | 4.74 | 3.20 | 4.65 | 38.14 | 87.14 |
| Gemini-2.0-Flash | 65.00 | 71.86 | 5.00 | 5.00 | 4.97 | 5.00 | 4.93 | 4.94 | 99.00 | 100.00 |
| Claude-3.5-Sonnet | 65.00 | 72.86 | 4.61 | 4.92 | 4.11 | 4.69 | 3.41 | 4.42 | 55.57 | 90.43 |
| LLaMA-3.3-70B | 65.71 | 72.43 | 4.66 | 4.87 | 4.37 | 4.76 | 4.34 | 4.80 | 66.00 | 88.29 |

(T1–T7), on four benchmark datasets generated by different agent families. The Base datasets represent the initial set of generated problems before difficulty scaling, while the Final datasets are the result of the subsequent Teacher-Student competition and Orchestrator validation processes, designed to increase the benchmark's difficulty.

Across all agent families and evaluation models, we observe a consistent drop in accuracy from the base to final benchmarks. This indicates that our protocol successfully increases task difficulty in a controlled manner. On average, evaluation accuracy drops by approximately 37.3 percentage points after the adaptive scaling phase, highlighting the non-trivial nature of the final problems. Importantly, despite the increased difficulty, the final problems maintain high quality, as validated separately (see Section 4.4). This substantial reduction in accuracy confirms that the competitive interaction between the Teacher and Student agents, coupled with the Orchestrator's validation, successfully led to the creation of more challenging anomaly detection instances.

## 4.4 ORCHESTRATOR VALIDATION

This section underscores the crucial role of the Orchestrator agent in ensuring the quality and validity of our text anomaly detection benchmark. To demonstrate this, we compared the performance of several LLMs on two versions of a benchmark generated by GPT-4o agents: one created solely through the Teacher-Student competition protocol (without an Orchestrator) and the other generated using our full framework, including Orchestrator validation. Table 3 presents this comparison, showing the evaluation performance of GPT-4o, Gemini-2.0-Flash, Claude-3.5-Sonnet, and LLaMA-3.3-70B on both benchmark versions.

At first glance, the benchmark generated without an Orchestrator appears more challenging—evaluation accuracy is consistently lower across all models. However, when we analyze problem quality along dimensions such as validity, coherence (logical consistency and type adherence), and fairness, we observe a notable degradation in quality. This suggests that the lower performance is not due to truly challenging reasoning tasks but rather to flawed or ambiguous question design. In other

words, the competitive protocol without validation tends to inflate difficulty artificially by generating problems that are confusing or ill-posed.

By contrast, our Orchestrator-guided pipeline maintains higher quality across all metrics while still increasing difficulty. The Orchestrator filters out problems that are ill-formed, inconsistent, or lack a clear solution, ensuring that performance drops are reflective of genuine reasoning challenges—not annotation noise or design failures. These findings emphasize the critical role of the Orchestrator in producing challenging yet fair benchmarks, where performance gaps more accurately reflect model capability rather than dataset artifacts.

### 4.5 SCENARIO: EVALUATING FUTURE LLM CAPABILITIES

To examine the sustainability of our benchmark under the rapid pace of LLM advancements, we simulate a future scenario where newer models outperform the current generation. Specifically, we assume GPT-4o as the generation model—serving as Teacher, Student, and Orchestrator—and evaluate the resulting benchmark using GPT-o3-mini and GPT-o4-mini, hypothetical successors representing future LLMs.

As shown in Table 4, all models—including the current GPT-4o—achieve near-ceiling accuracy on the base problems, highlighting the limitation of static benchmark design. However, when evaluated on the final benchmark constructed through our difficulty-scaling protocol, performance drops substantially for all models. Notably, GPT-o3-mini and GPT-o4-mini score lower than GPT-4o, despite being assumed as future improvements.

Table 4: Simulated future scenario with GPT-o3/o4-mini (future) vs. GPT-4o/4o-mini (current), showing sustained relative evaluation.

| Evaluation Model | GPT-4o | |
| --- | --- | --- |
| | Base | Final |
| GPT-o3-mini | 93.71 | 72.14 |
| GPT-o4-mini | 91.86 | 72.43 |
| GPT-4o | 94.29 | 72.43 |
| GPT-4o-mini | 93.00 | 68.29 |

This demonstrates that our benchmark not only scales difficulty in response to the generator's capability but also maintains long-term relevance. Unlike static benchmarks that saturate over time, our framework supports **relative evaluation**, where difficulty dynamically adapts to each generation model, allowing performance gaps between models to remain meaningful. Even as LLMs grow more powerful, our protocol preserves discriminative power—enabling robust comparison across models, regardless of when they are developed.

### 4.6 CONSISTENCY AND STABILITY IN BENCHMARK GENERATION

To ensure that our benchmark protocol supports not only adaptability but also **reliable reproducibility**, we evaluate the consistency of benchmark quality across repeated generations. In this experiment, we repeatedly generate benchmark datasets using the same agent configuration—Gemini-2.0-Flash as the Teacher, Student, and Orchestrator—and measure the performance of GPT-4o-mini, a representative model from a different family (GPT series), on these benchmarks. We generate 50 samples per task (350 in total) in the first round, then incrementally add 50 samples per task in each subsequent round, up to 1000 samples per task. For each round (50 to 1000 samples), we evaluate GPT-4o-mini on the corresponding benchmark and track its average accuracy across the seven anomaly detection tasks. Figure 4 plots model accuracy

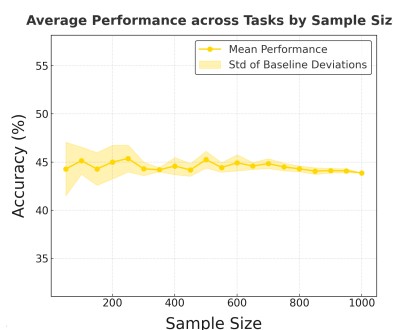

Figure 4: Consistency in Benchmark Generation.

per task as a function of the number of generated samples. We observe that performance remains largely stable across sample sizes, with only minor fluctuations. This result shows that our benchmark generation protocol is not only adaptive and dynamic, but also statistically stable across runs.

## 5 CONCLUSION

We present ATAD, an agent-centric benchmark protocol that adaptively generates and validates reasoning-focused anomaly detection tasks. By shifting from static datasets to dynamic protocols,

ATAD enables sustainable, scalable, and stable evaluation of ever-evolving LLMs. Our results demonstrate that ATAD surfaces reasoning failures missed by conventional benchmarks and enables model-benchmark co-evolution, offering actionable insights into model-specific reasoning gaps. Future work includes extending ATAD to track evolving LLMs and advancing text anomaly detection as a reasoning benchmark.

ACKNOWLEDGMENTS

The work of Junhyun Lee was supported by the Hankuk University of Foreign Studies Research Fund (2026).

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

## A APPENDIX OVERVIEW

In this appendix, we provide extended analyses and additional details to complement the main paper. Specifically, we include the following: (1) detailed performance results across benchmarks generated by four different models, expanding upon the averaged results in Table 1 of the main paper; (2) expanded descriptions of the seven anomaly detection task types, along with their associated topics and anomaly factors; (3) prompt templates and examples used for task generation and validation; (4) failure cases rejected by the Orchestrator during the benchmark validation phase; (5) performance details corresponding to Figure 4 of the main paper; (6) additional related work not included in the main text due to space constraints; (7) a model-family bias analysis using the introduced Bias Index; (8) an analysis of task-wise generation characteristics and constraints; (9) formal definitions and empirical validation of the difficulty scaling mechanism; (10) stability analysis of the protocol under heterogeneous agent configurations; (11) a conceptual comparison with prior dynamic benchmarking frameworks; and (12) future research directions for extending our protocol.

## B EVALUATION RESULTS BY BENCHMARK GENERATOR

In Table 5, we report detailed accuracy tables for each benchmark individually generated by GPT-4o, Gemini-2.0-Flash, Claude-3.5-Sonnet, and LLaMA-3.3-70B, complementing the averaged results shown in Table 1 of the main paper. Each table presents the performance of the twelve evaluation models on a benchmark created by one of the four generator models.

Interestingly, as shown in Table 5, there is no single evaluation model that consistently outperforms others across all benchmarks. Although one might expect GPT-family models to perform best on the benchmark generated by GPT-4o, we observe that Claude-3.5-Sonnet achieves the highest average score in that case, as reported in Table 5a. This suggests that the identity of the generator model does not systematically favor or disadvantage any particular evaluation model family. The observed performance differences are more attributable to the inherent difficulty and heterogeneity of the benchmarks, rather than to any systematic advantage conferred to evaluation models by alignment with the generator.

## C DESCRIPTIONS OF ANOMALY DETECTION TASK TYPES

This section provides additional details on the seven text anomaly detection tasks introduced in Section 3 of the main paper. Each task is designed to evaluate a distinct aspect of LLM reasoning, ranging from contextual and discourse coherence to ambiguity resolution and logical consistency.

As summarized in Table 6, we present the input and output formats for each task, reflecting how anomaly instances are structured and what form of prediction is expected from the model. These formats fall into three structural categories: (1) identifying an anomalous sentence within a paragraph (T1, T5, T6, T7), (2) selecting an inappropriate option from a given list of candidates (T3, T4), and (3) determining whether the overall sentence order in a paragraph is coherent (T2). The corresponding outputs are represented either as index selections or binary judgments, depending on the task type.

Beyond structural design, Table 7 outlines the core reasoning types targeted by each task, the specific challenge factors incorporated to enhance difficulty, and the domain topics used to amplify reasoning complexity. Challenge factors—such as subtle semantic deviations, logical reversals, or ambiguous pronouns—are selectively added to a subset of samples to increase difficulty while preserving clarity. To promote diversity and prevent overfitting to specific patterns, each factor is applied with a 50% probability during problem generation.

To ensure comprehensive coverage of academic reasoning, task content is curated across six high-level domains (e.g., science, economics, philosophy). Each task is paired with domains that naturally emphasize the relevant reasoning challenge, facilitating a principled topic-to-task alignment. This mapping is shown in the final column of Table 7.

While each task is designed around a primary reasoning capability, many demand compound reasoning skills—for example, Task T4 (Bridge Sentence Evaluation) requires not only logical coherence but also sensitivity to topic transitions. Such multifaceted design enables our benchmark to assess nuanced reasoning failures beyond surface-level understanding.

Table 5: Performance of evaluation models on benchmarks generated by GPT-4o, Gemini-2.0-Flash, Claude-3.5-Sonnet, and LLaMA-3.3-70B.

(a) Generated by GPT-4o

| Evaluation Model | T1 | T2 | T3 | T4 | T5 | T6 | T7 | Avg. |
|---|---|---|---|---|---|---|---|---|
| GPT-3.5-Turbo OpenAI (2024a) | 78.00 | 22.00 | **85.00** | 71.00 | 50.00 | 76.00 | 92.00 | 67.71 |
| GPT-4o-mini OpenAI (2024c) | 80.00 | 19.00 | 77.00 | 74.00 | 55.00 | 78.00 | 95.00 | 68.29 |
| GPT-4o OpenAI (2024b) | 84.00 | 22.00 | 80.00 | 81.00 | 59.00 | 86.00 | 95.00 | 72.43 |
| GPT-o4-mini OpenAI (2024d) | 84.00 | 30.00 | 82.00 | 79.00 | 54.00 | 82.00 | 96.00 | 72.43 |
| Gemini-1.5-Flash Reid et al. (2024) | 5.00 | 14.00 | 75.00 | 65.00 | 10.00 | 9.00 | 34.00 | 30.29 |
| Gemini-2.0-Flash-Lite Google DeepMind (2024) | 85.00 | 14.00 | 77.00 | 73.00 | **62.00** | 78.00 | 95.00 | 69.14 |
| Gemini-2.0-Flash Google DeepMind (2024) | 86.00 | 23.00 | 78.00 | 80.00 | 56.00 | 83.00 | 97.00 | 71.86 |
| Claude-3-Haiku Anthropic (2024b) | 80.00 | 14.00 | 78.00 | 77.00 | 52.00 | 84.00 | 90.00 | 67.86 |
| Claude-3.5-Haiku Anthropic (2024a) | 23.00 | **50.00** | 2.00 | 10.00 | 6.00 | 8.00 | 33.00 | 18.86 |
| Claude-3.5-Sonnet Anthropic (2024a) | **88.00** | 29.00 | 78.00 | 84.00 | 47.00 | 86.00 | **98.00** | **72.86** |
| LLaMA-3.1-8B Llama Team (2024) | 56.00 | 19.00 | 48.00 | 31.00 | 50.00 | 51.00 | 74.00 | 47.00 |
| LLaMA-3.3-70B Llama Team (2024) | 85.00 | 30.00 | 78.00 | **86.00** | 52.00 | 80.00 | 96.00 | 72.43 |

(b) Generated by Gemini-2.0-Flash

| Evaluation Model | T1 | T2 | T3 | T4 | T5 | T6 | T7 | Avg. |
|---|---|---|---|---|---|---|---|---|
| GPT-3.5-Turbo OpenAI (2024a) | 43.00 | 1.00 | 54.00 | 29.00 | 43.00 | 30.00 | 94.00 | 42.00 |
| GPT-4o-mini OpenAI (2024c) | 39.00 | 5.00 | 52.00 | **36.00** | 37.00 | 37.00 | 93.00 | 42.71 |
| GPT-4o OpenAI (2024b) | 44.00 | 6.00 | 59.00 | 33.00 | 42.00 | 34.00 | 95.00 | 44.71 |
| GPT-o4-mini OpenAI (2024d) | **48.00** | 15.00 | **62.00** | 32.00 | 39.00 | 41.00 | 93.00 | 47.14 |
| Gemini-1.5-Flash Reid et al. (2024) | 0.00 | 1.00 | 56.00 | 29.00 | 11.00 | 13.00 | 9.00 | 17.00 |
| Gemini-2.0-Flash-Lite Google DeepMind (2024) | 44.00 | 1.00 | 60.00 | 27.00 | **45.00** | **47.00** | 94.00 | 45.43 |
| Gemini-2.0-Flash Google DeepMind (2024) | 44.00 | 3.00 | 52.00 | 32.00 | 40.00 | 43.00 | **96.00** | 44.29 |
| Claude-3-Haiku Anthropic (2024b) | 47.00 | 2.00 | 47.00 | 32.00 | 37.00 | 45.00 | 90.00 | 42.86 |
| Claude-3.5-Haiku Anthropic (2024a) | 15.00 | **54.00** | 19.00 | 3.00 | 3.00 | 11.00 | 68.00 | 24.71 |
| Claude-3.5-Sonnet Anthropic (2024a) | 44.00 | 21.00 | 61.00 | 33.00 | 39.00 | 39.00 | 95.00 | **47.43** |
| LLaMA-3.1-8B Llama Team (2024) | 29.00 | 2.00 | 26.00 | 10.00 | 40.00 | 29.00 | 64.00 | 28.57 |
| LLaMA-3.3-70B Llama Team (2024) | 42.00 | 6.00 | 53.00 | 31.00 | 39.00 | 39.00 | 95.00 | 43.57 |

(c) Generated by Claude-3.5-Sonnet

| Evaluation Model | T1 | T2 | T3 | T4 | T5 | T6 | T7 | Avg. |
|---|---|---|---|---|---|---|---|---|
| GPT-3.5-Turbo OpenAI (2024a) | 67.00 | 24.00 | 69.00 | 72.00 | 58.00 | 54.00 | 86.00 | 61.43 |
| GPT-4o-mini OpenAI (2024c) | 61.00 | 14.00 | 59.00 | 75.00 | 55.00 | **57.00** | 84.00 | 57.86 |
| GPT-4o OpenAI (2024b) | **73.00** | 26.00 | 70.00 | 73.00 | 57.00 | 54.00 | 86.00 | 62.71 |
| GPT-o4-mini OpenAI (2024d) | 71.00 | 36.00 | **72.00** | 71.00 | 50.00 | 54.00 | 79.00 | 61.86 |
| Gemini-1.5-Flash Reid et al. (2024) | 8.00 | 13.00 | 60.00 | 75.00 | 20.00 | 2.00 | 20.00 | 28.29 |
| Gemini-2.0-Flash-Lite Google DeepMind (2024) | 67.00 | 8.00 | 63.00 | 74.00 | **62.00** | **57.00** | 81.00 | 58.86 |
| Gemini-2.0-Flash Google DeepMind (2024) | **73.00** | 18.00 | 70.00 | **78.00** | 48.00 | 55.00 | **91.00** | 61.86 |
| Claude-3-Haiku Anthropic (2024b) | 69.00 | 2.00 | 61.00 | 72.00 | 59.00 | 51.00 | 68.00 | 54.57 |
| Claude-3.5-Haiku Anthropic (2024a) | 21.00 | **60.00** | 2.00 | 4.00 | 3.00 | 12.00 | 28.00 | 18.57 |
| Claude-3.5-Sonnet Anthropic (2024a) | 71.00 | 26.00 | 64.00 | 78.00 | 58.00 | 55.00 | **91.00** | 63.29 |
| LLaMA-3.1-8B Llama Team (2024) | 22.00 | 16.00 | 29.00 | 42.00 | 44.00 | 24.00 | 58.00 | 33.57 |
| LLaMA-3.3-70B Llama Team (2024) | 67.00 | 45.00 | 62.00 | **78.00** | 59.00 | 53.00 | 88.00 | **64.57** |

(d) Generated by LLaMA-3.3-70B

| Evaluation Model | T1 | T2 | T3 | T4 | T5 | T6 | T7 | Avg. |
|---|---|---|---|---|---|---|---|---|
| GPT-3.5-Turbo OpenAI (2024a) | 48.00 | 17.00 | 59.00 | 22.00 | 72.00 | 47.00 | 54.00 | 45.57 |
| GPT-4o-mini OpenAI (2024c) | 49.00 | 30.00 | 62.00 | 31.00 | 63.00 | 63.00 | 60.00 | 51.14 |
| GPT-4o OpenAI (2024b) | 47.00 | 31.00 | 64.00 | 26.00 | 39.00 | 53.00 | 48.00 | 44.00 |
| GPT-o4-mini OpenAI (2024d) | 50.00 | 40.00 | 58.00 | 30.00 | 46.00 | 52.00 | 52.00 | 46.86 |
| Gemini-1.5-Flash Reid et al. (2024) | 11.00 | 17.00 | 57.00 | 26.00 | 29.00 | 10.00 | 21.00 | 25.71 |
| Gemini-2.0-Flash-Lite Google DeepMind (2024) | **60.00** | 20.00 | 54.00 | 35.00 | **82.00** | 66.00 | 75.00 | 56.00 |
| Gemini-2.0-Flash Google DeepMind (2024) | 58.00 | **56.00** | 52.00 | 43.00 | 60.00 | **67.00** | 68.00 | **57.71** |
| Claude-3-Haiku Anthropic (2024b) | 59.00 | 30.00 | 58.00 | 26.00 | 66.00 | 60.00 | 43.00 | 48.86 |
| Claude-3.5-Haiku Anthropic (2024a) | 20.00 | **56.00** | 6.00 | 3.00 | 10.00 | 3.00 | 13.00 | 15.86 |
| Claude-3.5-Sonnet Anthropic (2024a) | **60.00** | 51.00 | 57.00 | 43.00 | 70.00 | 50.00 | 63.00 | 56.29 |
| LLaMA-3.1-8B Llama Team (2024) | 51.00 | 14.00 | 39.00 | 15.00 | 78.00 | 51.00 | **79.00** | 46.71 |
| LLaMA-3.3-70B Llama Team (2024) | 49.00 | 30.00 | **60.00** | **45.00** | 59.00 | 59.00 | 58.00 | 51.43 |

# D   PROMPT TEMPLATES AND EXAMPLES

This section presents the prompt templates used in our benchmark pipeline. We categorize prompts into two primary roles: (1) generation prompts used by the **Teacher agent** to construct task instances, and (2) validation prompts used by the **Orchestrator agent** to assess the quality and structure of those instances.

The generation prompts are designed to be style-specific (e.g., GRE-style) and conditionally incorporate difficulty scaling instructions and challenge factors (e.g., semantic deviation, logical inconsistency). Each prompt guides the Teacher to generate one of the seven anomaly task types (T1–T7) in a consistent JSON schema.

Table 6: Input/output structure for each text anomaly detection task.

| Task ID | Task Name | Input | Output |
|---|---|---|---|
| T1 | Sentence Context Anomaly | 5–6 sentence paragraph | Index of off-topic sentence |
| T2 | Paragraph Order Consistency | 5-sentence paragraph | Boolean (True/False) |
| T3 | Blank-based Choice Anomaly | Sentence with blank + 5 choices | Index of most inappropriate choice |
| T4 | Bridge Sentence Evaluation | Two paragraphs + 5 bridge candidates | Index of incoherent bridge |
| T5 | Referential Ambiguity | 5-sentence paragraph | Index of ambiguous sentence |
| T6 | Logical Contradiction | 5-sentence paragraph | Index of logically inconsistent sentence |
| T7 | Tone / Style Violation | 5-sentence paragraph | Index of tone/style violation |

Table 7: Reasoning types, challenge factors, and domain topics per anomaly detection task.

| Task ID | Reasoning Type | Challenge Factors | Topics (Domains) |
|---|---|---|---|
| T1 | Contextual reasoning | Minor topic shift, semantic deviation | Philosophy, society, psychology |
| T2 | Discourse coherence | Sentence reordering | Science, economics, politics |
| T3 | Lexical + pragmatic reasoning | Lexical fit, collocation | Literature, psychology, philosophy |
| T4 | Logical bridging + topic shift detection | Weak logical connection, abrupt topic shift | Economics, society, policy |
| T5 | Coreference resolution | Ambiguous pronouns, unclear referents | Psychology, literature, philosophy |
| T6 | Causal and contradiction reasoning | Contradictory claims, causal reversal | Science, economics, politics |
| T7 | Stylistic reasoning | Tone shift, register mismatch | Literature, philosophy |

The validation prompts ensure that the generated problems are well-formed, solvable, and coherent. These prompts are used by the Orchestrator during three key phases of the protocol: (1) immediately after the Teacher generates an initial problem (Initial Validation); (2) after the Student solves the problem correctly, to provide feedback for generating a harder version (Feedback for Difficulty Escalation); (3) once a new, difficulty-scaled version of the problem is created, to ensure it maintains quality and appropriate challenge (Validation of Difficulty-Scaled Problem). Below, we present representative prompt examples for Task T1 across all three phases.

For clarity, we simplify the full prompt, and the full set of prompt templates used for these phases is available in the supplementary material.

## D.1 TEACHER PROMPT FOR TASK T1 (GENERATION)

You are a GRE-style exam question generator. Create a question for task T1 on the topic of psychology.
Generate 5 to 6 sentences on psychology. One of them should be anomalous (e.g., semantically inconsistent or conceptually off-topic).
The anomaly should be based on: semantic deviation.
Create a non-trivial anomaly that requires careful reading to detect. It should be noticeable but not immediately obvious.

Return the result strictly in JSON format:
```
    {
      "context": ["..."],
      "anomaly_index": <integer>,
      "meta": {
          "source": "GRE",
          "topic": "psychology",
          "anomaly_type": "semantic deviation"
      }
    }
```

## D.2 ORCHESTRATOR PROMPT FOR TASK T1 (VALIDATION)

### D.2.1 INITIAL VALIDATION PROMPT

You are a benchmark quality controller evaluating if this problem is well-formed and structured correctly for task T1.

Task Type: Sentence Context Anomaly (T1)

Task Description: This task requires generating 5–6 sentences on a topic where one of them is anomalous (semantically inconsistent or conceptually off-topic). The anomaly should be detectable but not overly obvious, requiring careful reading to identify.

Expected Structure:
- "context": array of 5–6 sentences
- "anomaly_index": integer indicating the anomalous sentence
- "meta": source, topic, anomaly_type

Context:
1. Sentence A
2. Sentence B
...
5. Sentence E

Correct Answer: Option 4

Evaluate the problem based on these criteria:
1. VALIDITY: Is the problem well-formed and complete?
2. TYPE ADHERENCE: Does the problem follow the expected task type requirements?
3. LOGICAL COHERENCE: Is the anomaly identifiable?
4. FAIRNESS: Is the problem fair and reasonable? Does it have a clear, unambiguous solution?

Return your evaluation in JSON format:
```
{
    "approved": boolean (true if the problem passes all criteria, false otherwise),
    "feedback": null if approved, or detailed feedback if rejected addressing:
            - Problem construction issues
            - Anomaly ambiguity concerns
            - Specific improvement suggestions
}
```

### D.2.2 FEEDBACK PROMPT (ORCHESTRATOR TO TEACHER) FOR DIFFICULTY ESCALATION

When a Student successfully solves a task, the Orchestrator analyzes the Student's explanation and provides structured feedback to help the Teacher generate a harder version of the problem. This feedback prompt includes the original problem, the student's reasoning, and a checklist to guide difficulty escalation. Below is the full prompt used for this purpose.

You are helping to create a harder version of a problem that a student has correctly solved. Analyze the student's solution and provide feedback.

Task Type: Sentence Context Anomaly (T1)
Current Difficulty: easy

ORIGINAL PROBLEM:
```
{
    "context": [
        "Cognitive dissonance occurs when individuals experience conflicting beliefs.",
        "It can cause discomfort and lead to attitude change.",
        "Festinger's theory explains how people resolve dissonance.",
        "Photosynthesis is the process by which plants convert light into energy.",
        "Dissonance reduction strategies include rationalization and denial."
    ],
    "anomaly_index": 3,
    "meta": {
        "source": "GRE",
        "topic": "psychology",
        "anomaly_type": "semantic deviation",
        "difficulty": "easy"
    }
}
```

Student's Explanation: "The sentence about photosynthesis is unrelated to the other sentences on cognitive dissonance. It's a semantic outlier."

Based on how the student solved this problem, provide feedback to create a more challenging version:
1. What aspects did the student easily identify?
2. How could the problem be made more subtle or complex?
3. Give specific suggestions for increasing difficulty.

Return your feedback in JSON format:
```
{
    "analysis": "Brief analysis of student solution",
    "suggestions": ["Specific suggestion 1", "Specific suggestion 2", ...],
    "difficulty_increase": "Summary of how to increase difficulty"
}
```

### D.2.3 VALIDATION OF DIFFICULTY-SCALED PROBLEM

After the Teacher generates a more difficult version of the original problem, the Orchestrator evaluates its quality to determine whether the sample meets the necessary criteria for inclusion in the benchmark. This prompt includes a task-specific description, the expected output structure, the difficulty level, and the sample content. The Orchestrator then assesses whether the problem is well-formed, challenging, and coherent. Below is the full prompt used in this validation phase for Task T1.

---

You are a benchmark quality controller evaluating if a problem with increased difficulty is well-formed and appropriate for task T1.

Task Type: Sentence Context Anomaly (T1)
Difficulty Level: hard

Task Description: This task requires generating 5–6 sentences on a topic where one of them is anomalous (semantically inconsistent or conceptually off-topic). The anomaly should be detectable but not overly obvious, requiring careful reading to identify.

Expected Structure: The expected JSON structure should include 'context' (array of 5-6 sentences), 'anomaly_index' (integer indicating which sentence is anomalous), and 'meta' (with source, topic, and anomaly_type).

Context:
1. Cognitive dissonance occurs when individuals experience conflicting beliefs.
2. It can cause discomfort and lead to attitude change.
3. Festinger's theory explains how people resolve dissonance.
4. Social conformity often influences decision-making in groups.
5. Dissonance reduction strategies include rationalization and denial.
6. A dissonance-free state enhances psychological consistency.

Correct Answer: Option 4

Note: While maintaining quality standards, be lenient in your evaluation. Accept problems that are reasonable and solvable, even if they have minor imperfections.

Evaluate the problem based on these criteria:
1. VALIDITY: Is the problem well-formed and complete?
2. TYPE ADHERENCE: Does the problem follow the expected task type requirements?
3. LOGICAL COHERENCE: Is the correct answer clearly identifiable?
4. FAIRNESS: Is the problem fair and reasonable? Does it have a clear, unambiguous solution?
5. DIFFICULTY: Is the difficulty appropriate for hard level?

Return your evaluation in JSON format:
    {
        "approved": boolean (true if the problem passes all criteria, false otherwise),
        "feedback": null if approved, or detailed feedback if rejected addressing:
                    - Problem construction issues
                    - Anomaly ambiguity concerns
                    - Difficulty appropriateness
                    - Specific improvement suggestions
    }

---

## E    REJECTED CASES FROM THE ORCHESTRATOR VALIDATION

We present two distinct analyses to illustrate the role of the Orchestrator in problem validation. Section E.1 examines how rejections lead to improved samples by comparing rejected and subsequently approved versions. Section E.2 explores the consequences of using format-compliant but semantically flawed problems that were rejected, showing how such issues can affect model performance when the Orchestrator is removed.

### E.1    REFINED AFTER REJECTION

One of the key functions of the Orchestrator is not just to detect flawed problems but to guide their improvement. Figure 5 illustrates a representative case from the T1 task (Sentence Context Anomaly), where the problem was rejected for lacking a clear anomaly and subsequently revised into a higher-quality version.

In the rejected version (left), all five sentences are factually correct and topically coherent, making it difficult to identify a distinct anomaly. The fifth sentence about Kant's influence on epistemology, while slightly tangential, remains within the bounds of acceptable variation in context. The Orchestrator flagged this problem as ill-suited for high-difficulty evaluation due to the lack of a clear semantic deviation.

After feedback, the Teacher produced a revised version (right) on existentialist philosophy, where the anomaly subtly introduces scientifically framed misinformation: it claims Camus's absurdism was based on quantum mechanics, which is factually incorrect. This revised problem is more appropriate for an "extreme" difficulty level as it requires nuanced understanding of philosophical context to detect the inconsistency.

This example demonstrates how Orchestrator feedback can elevate problem quality by transforming ambiguous or unfocused items into more challenging and pedagogically valid benchmark samples.

### E.2    STRUCTURALLY VALID BUT SEMANTICALLY FLAWED

This example highlights the importance of Orchestrator validation even when the problem format adheres to the expected task type. While structural correctness (e.g., number of options, sentence layout) can be verified without the Orchestrator, semantic soundness often cannot.

The problem shown here follows the T3 task format correctly but was rejected due to an unclear anomaly. Both "literary impressionism" and "hyperrealism" are loosely connected to the context, making the intended anomaly debatable.

Without validation, such problems could remain in the benchmark and appear reliable—yet when tested on three strong LLMs, two of them (GPT-4o and Claude 3.5 Sonnet) failed to answer correctly. This illustrates that structurally valid but semantically underspecified problems can mislead evaluation outcomes.

While this can happen in any phase, it becomes especially risky during difficulty escalation, where the Teacher agent may try to make problems harder but instead make them ambiguous.

## F    PERFORMANCE DETAILS FOR CONSISTENCY FIGURE

To support the results presented in Figure 4 of the main paper, this section provides additional details on how the consistency plot was computed. The figure tracks the average accuracy of GPT-4o-mini across different sample sizes—ranging from 50 to 1000 samples per task (T1–T7).

For each sample size, we evaluate the model's accuracy per task and calculate the average across tasks. The shaded region around the curve represents the standard deviation of task-wise deviations from the final round (i.e., the 1000-sample benchmark). For each sample size, we measure how much each task's accuracy differs from its corresponding value at 1000 samples, and compute the standard deviation across these deviations. This provides a straightforward view of consistency over time, showing that performance remains stable across all sample sizes—not just at the final stage.

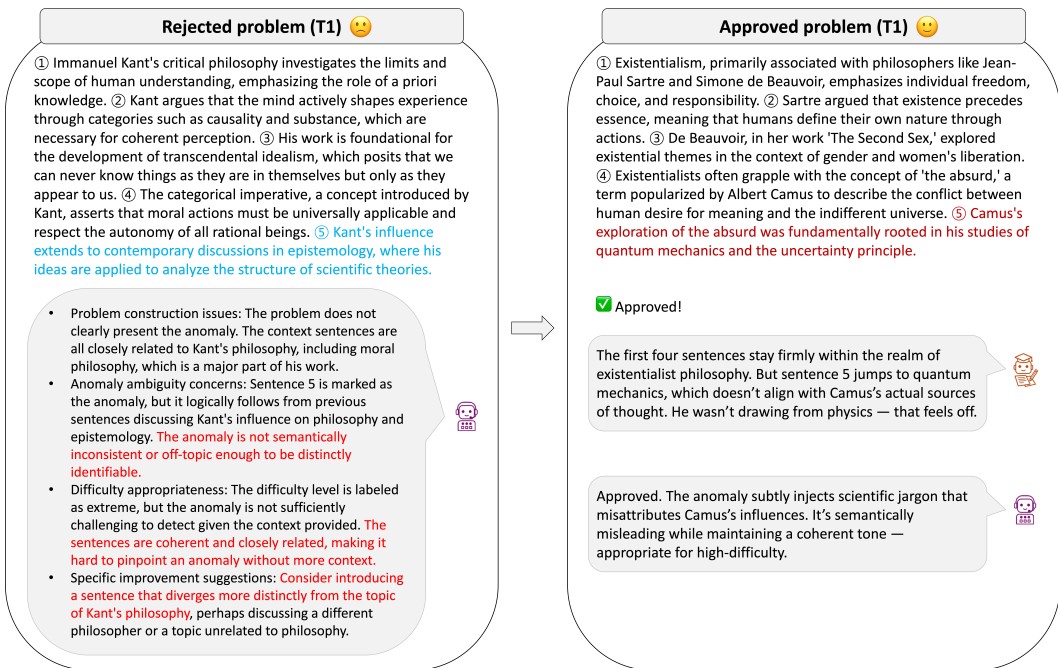

Figure 5: **Refined after rejection.** The left side shows a rejected T1 (Sentence Context Anomaly) problem where the anomaly was conceptually weak and difficult to identify. The Orchestrator's feedback noted the lack of semantic inconsistency and suggested stronger topic divergence. The revised version (right) introduces a scientifically framed yet incorrect statement about Camus's influences, resulting in a clearer and more pedagogically effective anomaly. This highlights the Orchestrator's role in guiding high-difficulty problem construction.

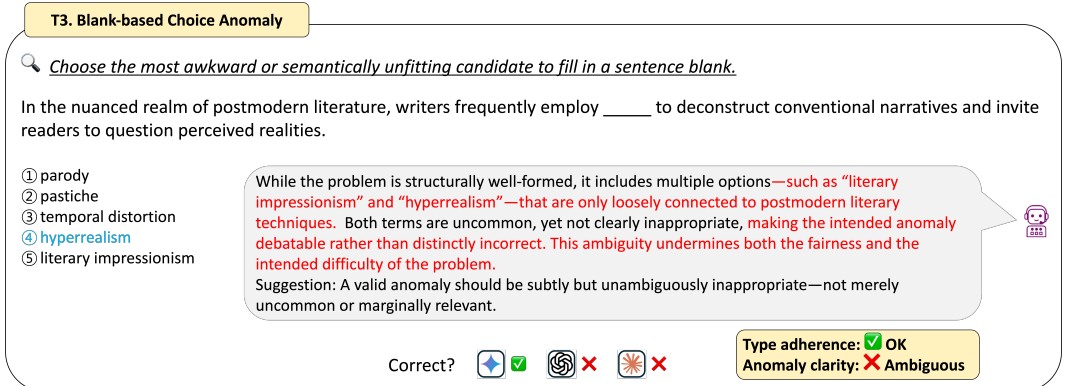

Figure 6: A structurally valid T3 problem rejected by the Orchestrator due to an unclear anomaly. Despite adhering to the task format, the anomaly is too ambiguous—leading two out of three LLMs to answer incorrectly.

As shown in the figure, the accuracy remains largely stable with only minor variations, indicating the robustness and consistency of our benchmark generation process.

# G    RELATED WORK

## G.1    TEXT ANOMALY DETECTION BENCHMARKS

Recent benchmarks have begun explicitly evaluating an LLM's ability to detect linguistic anomalies and coherence breaks in text. For example, CoheSentia Maimon & Tsarfaty (2023) introduces a

human-annotated coherence dataset of 500 AI-generated paragraphs, with both holistic and incremental (sentence-by-sentence) coherence scores. DECOR Zhang et al. (2024a) focuses on incoherence in L2 English writing, providing expert-labeled context–sentence pairs for detecting coherence breaks, explaining their causes (e.g. lack of cohesion or consistency), and even rewriting the incoherent sentences. Disco-Bench Wang et al. (2023) targets discourse-level anomalies by evaluating model performance on document-level test suites rich in cohesion and coherence phenomena across multiple tasks. Other well-known "anomaly" challenges include the Adversarial NLI dataset (ANLI)Nie et al. (2020), collected in three rounds of human-and-model-in-the-loop adversarial examples, and the Winograd Schema ChallengeLevesque et al. (2012) for commonsense pronoun disambiguation. These benchmarks share a focus on uncovering subtle inconsistencies or incoherence in text. However, they are inherently limited by their static, human-curated nature. Each new example often requires costly human creativity and annotation, making it difficult to sustainably scale up the dataset or progressively increase task difficulty. ANLI, for instance, achieved increasing complexity over three rounds, but this required extensive human involvement at each round. In general, static anomaly datasets incur high labeling costs and quickly saturate—once models learn to solve the fixed set of examples, evaluation stagnates, and creating harder cases demands significant manual effort. This motivates exploration of more dynamic and automated evaluation protocols for textual coherence and anomaly detection.

## G.2 STATIC LLM EVALUATION BENCHMARKS

The standard paradigm for evaluating LLMs has been through fixed benchmarks covering a wide range of tasks. Notable examples include MMLU (Massive Multitask Language Understanding)Hendrycks et al. (2021), a 57-task exam covering diverse knowledge domains, GSM8K for grade-school math word problemsCobbe et al. (2021), and BIG-Bench Srivastava et al. (2023), a crowd-sourced collection of over 200 tasks probing various aspects of intelligence. These static benchmarks were initially effective for comparing models, but top-tier LLMs have rapidly saturated many of them. Models like GPT-4 now exceed or approach human-level performance on MMLU and GSM8K, leaving little headroom for differentiation. Moreover, concerns have arisen about *training data contamination*: since the evaluation sets are public and relatively small, powerful LLMs often inadvertently memorize or see similar questions during pre-training. This can inflate their scores without reflecting true reasoning progress, as evidenced by significant performance drops on rephrased or decontaminated test samples Shi et al. (2023). In short, static benchmarks are increasingly subject to memorization and ceiling effects. They also struggle to track evolving capabilities—once a benchmark is "solved" by current models, it cannot capture further improvements or new emergent reasoning skills. Beyond these, other important static benchmarks exist, such as AgentBench Liu et al. (2023), VisualAgentBench Liu et al. (2024), GAIA Mialon et al. (2023), ToolBench Qin et al. (2023), and HumanEval Chen et al. (2021), which primarily focus on isolated reasoning and generation capabilities of single agents, thereby failing to capture the intrinsic dynamics of multi-agent interactions. Recent efforts have proposed ever harder test sets (e.g. "MMLU 2.0" variants) and meticulous data filtering to mitigate leakage, but these are stopgap solutions. The inability of static evaluations to adapt alongside model progress motivates developing dynamic benchmarks that can continually pose fresh, unsolved challenges.

## G.3 DYNAMIC BENCHMARKS WITHOUT AGENTS

A growing line of work aims to generate evaluation data *dynamically* – creating new test samples on the fly to match a model's ability – without relying on multi-agent interactions. DyVal Zhu et al. (2024a) pioneered this approach with a general framework to algorithmically spawn new reasoning problems of controlled complexity. In DyVal, instead of a fixed dataset, a generation function produces test samples and a constraint mechanism modulates their complexity and validity in real time. One instantiation uses directed acyclic graphs to compose simple components into increasingly complex problems (e.g. multi-step math or logic puzzles), allowing systematic scaling of difficultyZhu et al. (2024a). These graph-based generated tasks require genuine reasoning and cannot be solved by mere memorization, but DyVal's template-driven nature limits it to certain domains (math, logical puzzles, algorithms). DARG (Dynamic LLM Evaluation via Adaptive Reasoning Graph)Zhang et al. (2024b) extends this idea by extracting the underlying reasoning graph of an existing benchmark problem and perturbing it to generate novel but related test samples. This yields new questions with tunable complexity levels while preserving coherence with the original data

distributionZhang et al. (2024b). Crucially, DARG uses an automated verifier (a code-augmented LLM) to ensure each generated sample's label or answer remains correct after perturbation, providing stronger ground truth guarantees. Broadly, these non-agent dynamic benchmarks demonstrate the ability to continuously adjust task difficulty and mitigate data contamination. Their main limitations lie in generality and validation: methods like DyVal rely on hand-crafted generation schemas (e.g. DAG operations) that are task-specific, while purely LLM-based generators risk producing invalid or trivial questions without additional checking. Ensuring robust quality control often requires an auxiliary procedure (such as DARG's code executor or heuristic filters) given the absence of a human or agent "referee." Thus, dynamic sample generation has shown promise in maintaining evaluation challenge, but incorporating more general and trustworthy validation remains an open challenge.

## G.4 Agent-Based Dynamic Evaluation Frameworks

Recent approaches have started to leverage AI *agents* (LLMs themselves) to both generate new evaluation items and verify their quality, yielding self-refreshing benchmarks. These can be grouped by the role agents play:

### G.4.1 Agent-Based Problem Verification

Several frameworks employ one or more LLM agents as internal *judges* or *verifiers* to ensure evaluation data quality and correctness. Benchmark Self-Evolving Wang et al. (2025) is a multi-agent system that iteratively refines existing benchmark questions: one agent perturbs the context or question (e.g. paraphrasing, adding noise or constraints) to make a new test instance, and another agent (or the model itself) attempts to solve it to verify that the instance is valid and non-trivial. By applying a set of such automated "reframing" operations, the benchmark can evolve dynamically with minimal human input. JudgeLM Zhu et al. (2025b) demonstrates that an LLM fine-tuned as a specialized evaluator can reliably score or check open-ended answers with >90% agreement to human judgment, effectively serving as a scalable replacement for human evaluation. This idea of an LLM-as-judge is also used in many dynamic benchmarks to replace costly human verification: for example, DyVal's follow-up work introduces "meta-probing agents" that automatically generate and check new reasoning challenges Zhu et al. (2024b), and the DARG framework's pipeline employs a code-execution agent to validate each generated sample's solution Zhang et al. (2024b). The BenchAgents system Butt et al. (2024) goes even further in modularizing the process: it deploys separate LLM agents for planning what data to create, for actually generating candidate problems, for verifying the correctness/quality of each candidate, and finally for assembling the evaluation and scoring models on it. By having agents explicitly double-check answers or filter out flawed questions, these frameworks instill a degree of robustness into dynamically created benchmarks. The verification agents can catch mistakes or ambiguities that a single-pass generation might miss, ensuring that the evolving evaluation data remains challenging yet fair. A downside, however, is that the agents themselves (being imperfect LLMs) might introduce their own biases or occasional errors in judgment, so careful design and calibration of the "judge" agents is required to maintain reliability. Beyond general LLM evaluation, specific frameworks like PersonaGym Samuel et al. (2024) have emerged for assessing specialized agents, introducing the first dynamic evaluation framework and automated human-aligned metric (PersonaScore) for persona agents, which are LLM agents designed to act according to an assigned persona.

### G.4.2 Problem Generation via Multi-Agent Protocols

Other works explore multi-agent *interaction protocols*—such as collaboration, debate, or competition— to automatically generate or evaluate content in novel ways. ChatEval Chan et al. (2024) is a representative example where multiple LLM-based critics form a "referee team" that debates and deliberates on the quality of a model's answer. By pitting several AI evaluators with different perspectives against each other in discussion, the evaluation becomes more robust than a single model's score, and the process mimics how multiple human judges arrive at a consensus. This multi-agent debate approach focuses on jointly evaluating content rather than generating new problems, but it showcases how agent interactions can replace and even surpass traditional human evaluation. In terms of content *generation*, BenchAgents (mentioned above) explicitly uses agents that cooperate (with minimal human oversight) to produce entirely new benchmark datasets — effectively automating the benchmark creation process through agent teamworkButt et al. (2024). There are also emerging

benchmarks to test the capabilities of multi-agent systems themselves. MultiAgentBenchZhu et al. (2025a) evaluates how well LLM agents can collaborate or compete in shared environments and tasks, introducing scenarios where multiple agents communicate to solve a problem. Its contribution is a suite of multi-agent challenge tasks (with coordination protocols like star or graph networks and metrics for teamwork quality), rather than an evolving benchmark, but it underscores the interest in agent-agent interaction. MeeseeksWang (2025) takes an iterative multi-turn approach: it simulates a realistic user interacting with an LLM by providing feedback on failed requirements, and measures whether the LLM can self-correct over multiple rounds. While not explicitly framed as multi-agent (the "user" feedback could be programmatic), it creates a feedback loop akin to two agents – one posing and refining the request, the other improving its answers – working together to achieve a correct solution. Many of these multi-agent or multi-turn frameworks successfully generate complex, rich interactions, but notably, most lack an explicit adversarial or difficulty-raising dynamic. Agents often cooperate to improve quality (as in collaborative problem solving or debate), rather than engaging in competitive play where one tries to stump or outpace the other. Likewise, the tasks or evaluations are usually predefined or randomly sampled rather than *progressively ramped up* in response to a model's mastery. For example, ChatEval's agents are not trying to make the task harder – they are jointly judging a given response. MultiAgentBench provides diverse scenarios but does not adapt scenario difficulty based on performance. In short, current multi-agent evaluation protocols focus on novel ways to assess or create content (often leveraging the wisdom of multiple AI judges or creators), yet they stop short of introducing a competitive teacher–student dynamic or automated curriculum that continuously pushes the model to its limit.

## H    MODEL-FAMILY BIAS ANALYSIS (BIAS INDEX)

To quantitatively assess whether ATAD embeds model-family-specific advantages, we introduce a simple Bias Index that measures whether models perform better on benchmarks generated by LLMs from the same family.

Let:

- G = generator family (GPT, Claude, Gemini, LLaMA)
- $M_{\text{same}}(G)$ = set of models belonging to family G
- $M_{\text{diff}}(G)$ = all evaluated models not belonging to family G

Given a benchmark generated by family G, we compute:

$$\text{BiasIndex}(G) = \text{MeanAccuracy}(M_{\text{same}}(G) \mid G) - \text{MeanAccuracy}(M_{\text{diff}}(G) \mid G).$$

A positive value would indicate a same-family advantage; a negative value would indicate a disadvantage. Values near zero indicate no family-specific bias.

Using the cross-family generation results from Appendix Table 5, we compute the Bias Index for all four generator families:

| Generator Family | BiasIndex | Highest-Accuracy Model |
|---|---|---|
| GPT | -0.53 | Claude-3.5-Sonnet |
| Gemini | -0.24 | Claude-3.5-Sonnet |
| Claude | 2.00 | LLaMA-3.3-70B |
| LLaMA | -0.57 | Gemini-2.0-Flash |

Table 8: BiasIndex across generator model families.

Across all families, BiasIndex(G) $\approx 0$, confirming that no model consistently performs better on benchmarks generated by its own family. In fact, none of the four families achieved the highest accuracy on a benchmark generated by themselves (e.g., GPT $\rightarrow$ Claude highest; Claude $\rightarrow$ LLaMA highest; Gemini $\rightarrow$ Claude highest; LLaMA $\rightarrow$ Gemini highest). This provides further evidence that ATAD's Orchestrator effectively removes surface-level stylistic cues or generator-specific artifacts, preventing any family-aligned advantage.

# I  TASK-WISE GENERATION CHARACTERISTICS

Different anomaly types in ATAD exhibit distinct generation characteristics that can influence both the number of validation cycles and the associated API cost. These variations arise from structural properties inherent to each task.

T1 (Sentence Context Anomaly) and T3 (Blank-based Choice Anomaly) show relatively stable generation behavior, as semantic-deviation and lexical-fit anomalies can be produced with low ambiguity.

T4 (Bridge Sentence Evaluation) and T6 (Logical Contradiction) are substantially more demanding to generate. Producing plausible yet subtly incorrect logical links or causal relationships requires finer control, leading to a higher rate of candidate rejection.

T5 (Referential Ambiguity) and T7 (Tone/Style Violation) also exhibit increased generation difficulty due to stricter requirements on pronoun/reference clarity and stylistic register consistency.

These task-specific characteristics explain why generation cost may vary across anomaly types, even under the same model and pricing scheme.

# J  FORMALIZATION AND EMPIRICAL VALIDATION OF DIFFICULTY

## J.1  FORMAL DEFINITION OF DIFFICULTY

In ATAD, we formally define the difficulty of a problem instance $q$ with respect to a Student model $S$ as the probability of failure conditioned on the problem being validated by the Orchestrator:

$$\text{Difficulty}(q; S) = \Pr[S \text{ fails on } q \mid q \in \text{Valid}], \tag{1}$$

where Valid denotes the set of problems that pass the Orchestrator's validation criteria, including well-formedness, logical coherence, task adherence, clarity, and fairness. This conditioning is essential: ATAD never interprets Student failures on defective, ambiguous, or ill-posed items as meaningful difficulty.

The probability is defined over the Student's stochastic decoding behavior. In practice, we estimate this quantity using the observed frequency of Student failures over Orchestrator-validated instances within each difficulty tier, where each instance is evaluated with a single forward pass.

$$\text{Difficulty}(D_t, S) = 1 - \text{Accuracy}(D_t, S), \tag{2}$$

where $D_t$ denotes a problem subset at difficulty tier $t$. Under this formulation, difficulty is a behaviorally grounded, solver-relative quantity: a problem is difficult precisely when a Student model of a given capability fails under the protocol's validity constraints.

This perspective is consistent with classical formulations in educational measurement and psychometrics, where item difficulty is defined via the probability of a correct (or incorrect) response for an examinee at a given proficiency level. Unlike latent-trait estimation in item response theory, however, ATAD directly observes failure probabilities under controlled validation, making difficulty operationally and empirically measurable for LLMs.

## J.2  DIFFICULTY TIERS ALONG A SINGLE GENERATION TRAJECTORY

ATAD does not assign difficulty labels heuristically or post-hoc. Instead, difficulty tiers emerge naturally along a single adversarial generation trajectory:

- The Teacher generates a base (Easy) problem.
- If the Student succeeds, the Teacher generates a more challenging variant.
- This process continues through progressively refined versions (Hard, Extreme).

Table 9: Average accuracy (%) across evaluation models as a function of difficulty tier. Each row corresponds to a benchmark generated under a different agent configuration. Difficulty tiers emerge along a single generation trajectory (Easy → Hard → Extreme → Impossible).

| Generator (Teacher / Student / Orchestrator) | Easy | Hard | Extreme | Impossible |
|---|---|---|---|---|
| GPT-4o family (4o / 4o / 4o) | 83.94 | 83.33 | 80.71 | 70.94 |
| Gemini-2.0-Flash family (2.0F / 2.0F / 2.0F) | 81.67 | 76.98 | 74.44 | 60.32 |
| Cross-family (Claude-4 / Gemini-2-Flash / GPT-4o) | 80.54 | 74.31 | 70.26 | 67.90 |

- For each generation trajectory, ATAD progressively produces Easy, Hard, Extreme, and Impossible variants of the same underlying problem. All tier-specific instances along the trajectory are retained and form the tiered benchmark subsets.

Thus, Easy, Hard, Extreme, and Impossible are not independently sampled problems, but correspond to distinct stopping points along the *same problem lineage*. This design ensures that increases in difficulty reflect genuine refinement of the same underlying reasoning challenge rather than distributional shifts across unrelated samples.

In the experiments reported in this section, we analyze only those trajectories that progressed through the full Impossible tier in order to allow consistent tier-wise comparison across generation regimes.

## J.3 EMPIRICAL MONOTONICITY ACROSS DIFFICULTY TIERS

To empirically validate the proposed definition, we evaluate three independent benchmark generation regimes:

1. Same-family GPT-4o agents (Teacher = Student = Orchestrator = GPT-4o),
2. Same-family Gemini-2.0-Flash agents (Teacher = Student = Orchestrator = Gemini-2.0-Flash),
3. Cross-family agents (Teacher = Claude-4-Sonnet, Orchestrator = GPT-4o, Student = Gemini-2.0-Flash).

For each regime, we compute the average accuracy across all evaluation models (GPT, Gemini, Claude, and LLaMA families) over tasks T1–T7 at each difficulty tier.

Across all three generation regimes, we observe a clear monotonic decrease in average accuracy as the difficulty tier increases from Easy to Hard, Extreme, and Impossible. This empirical trend directly validates our formal definition of difficulty as the Student's failure probability on Orchestrator-validated instances.

The same monotonic degradation is observed not only within each single-family generation regime but also consistently across different generation regimes. This indicates that the induced difficulty is not tied to a particular generator configuration, but reflects a robust, model-agnostic increase in reasoning complexity.

## J.4 IS ATAD DIFFICULTY DRIVEN BY GENUINE REASONING OR NARROW BLIND SPOTS?

A key concern in dynamic benchmark generation is whether increasing difficulty reflects genuine reasoning challenges or merely exploits narrow, idiosyncratic blind spots of the Student model used during generation. ATAD explicitly addresses this risk through both architectural design and empirical validation.

**Reasoning-conditioned difficulty escalation.** In ATAD, the Teacher does not blindly increase difficulty based solely on whether the Student succeeds or fails. Instead, the Orchestrator analyzes the Student's explanation and provides structured feedback to the Teacher, identifying the precise reasoning patterns that led to success or error. The Teacher is then instructed to refine the same underlying anomaly to make it more subtle or logically demanding, rather than introducing unrelated obscure knowledge or spurious topic shifts. Moreover, the Orchestrator enforces strict validation

rules on clarity, coherence, and fairness, rejecting items that rely on ambiguity or ill-defined cues. As a result, Student failures are used as difficulty signals only when the problem is a valid, unambiguous reasoning instance.

**Cross-family transfer and absence of self-preference.**  If difficulty were driven primarily by narrow blind spots of a particular Student model, one would expect benchmarks generated by a given family to become trivially easy for other models. However, the consistent difficulty trends observed across the three generation regimes in Table 9 indicate that the induced difficulty remains consistently non-trivial across different model families rather than collapsing outside the generator configuration.

We further observe no systematic self-preference phenomenon: models from the same family as the generator do not consistently outperform other families on their own generated benchmarks (see Table 5). For example, on the GPT-4o-generated benchmark, the best-performing model is often from a different family. This supports the conclusion that ATAD does not overfit to the reasoning idiosyncrasies of a specific Teacher–Student pair.

Taken together, the reasoning-conditioned escalation mechanism, the monotonic tier-wise degradation, and the absence of systematic self-preference provide converging evidence that ATAD difficulty reflects robust, generalizable reasoning demands rather than narrow model-specific artifacts.

## K  EFFECT OF USING THE SAME VS. DIFFERENT MODELS ACROSS AGENT ROLES

To analyze the effect of model homogeneity versus heterogeneity across agent roles, we conduct an additional experiment using a *cross-family configuration*, in which the Teacher, Student, and Orchestrator are instantiated with different LLM families (Teacher = Claude-4-Sonnet, Student = Gemini-2.0-Flash, Orchestrator = GPT-4o). This setting is compared against the standard *same-family configuration* used throughout the main paper (Teacher = Student = Orchestrator).

Table 10 reports the average accuracy of evaluation models under both configurations.

Table 10: Comparison between same-family and cross-family agent configurations.

| LLMs | Same-family | Cross-family |
|---|---|---|
| gpt-3.5-turbo | 54.18 | 51.71 |
| gpt-4o-mini | 55.00 | 53.86 |
| gpt-4o | 55.96 | 57.57 |
| gemini-2.0-flash-lite | 57.36 | 56.43 |
| gemini-2.0-flash | 58.93 | 54.57 |
| claude-3-haiku | 53.54 | 52.57 |
| claude-3.5-haiku | 19.50 | 19.57 |

The overall performance ordering and difficulty level are broadly preserved across the two configurations. Although absolute values differ slightly, the heterogeneous setting does not induce systematic changes in model ranking or benchmark difficulty under the same evaluation protocol.

These observations indicate that ATAD's behavior is primarily governed by the **protocol structure**—namely, the Teacher–Student adversarial loop together with Orchestrator-enforced validation—rather than by whether all agents originate from a single model family. By enforcing validity, clarity, and coherence, the Orchestrator mitigates stylistic or family-specific artifacts, which stabilizes the benchmark even when generation styles differ substantially across agents.

Taken together, these results suggest that ATAD remains stable under heterogeneous agent configurations. A broader exploration of additional cross-family combinations is left for future investigation.

Table 11: Conceptual comparison between ATAD and prior dynamic benchmark frameworks.

| Criterion | DyVal | DARG | Benchmark Self-Evolving | ATAD (Ours) |
|---|---|---|---|---|
| Domain | Math, logic | Graph augmentation | Benchmark refresh | **Text anomaly reasoning** |
| Generation Architecture | Generator + Filter | Generator + Critic | Generator + Selector | **Teacher → Orchestrator → Student** |
| Role Separation | ✗ Single loop | ✗ Mostly 2-agent | ✗ No 3-agent | **✓ 3-agent separation** |
| Difficulty Scaling | Static | No Student-conditioned loop | No difficulty conditioning | **✓ Student-conditioned escalation** |
| Validation Mechanism | Basic filtering | Code verifier | Quality scoring | **✓ Strict Orchestrator validation** |

## L    COMPARISON TO PRIOR DYNAMIC EVALUATION FRAMEWORKS (DYVAL, DARG, BENCHMARK SELF-EVOLVING)

DyVal, DARG, and Benchmark Self-Evolving represent influential efforts toward automated benchmark construction and renewal. However, their design objectives, target domains, and operational mechanisms differ fundamentally from those of ATAD. A structured comparison is provided in Table 11.

A central distinction lies in the level at which adaptation is performed. Existing frameworks operate primarily at the *data or graph level*, focusing on dataset extension, perturbation, or refresh. In contrast, ATAD is defined at the *protocol level*, where difficulty is adaptively regulated through an explicit Teacher–Orchestrator–Student interaction loop. Specifically, ATAD uniquely combines three elements that are absent in prior work: (i) an independent Orchestrator that enforces logical validity and clarity, (ii) Student-conditioned difficulty escalation driven by observed failure patterns, and (iii) an explicit three-agent role separation designed to expose fine-grained reasoning weaknesses. DyVal, DARG, and Benchmark Self-Evolving do not incorporate such failure-driven, protocol-level adversarial scaling mechanisms. Because prior frameworks are designed to modify or regenerate static benchmarks offline and subsequently evaluate models on fixed outputs, their objectives are not directly aligned with ATAD's adaptive difficulty-scaling paradigm. As a result, direct quantitative performance comparisons would not be meaningful under their respective design assumptions.

ATAD should therefore be viewed not as a data-generation technique, but as a **protocol-level difficulty regulation framework**. Rather than refreshing datasets, it enables the continual regeneration of adversarial reasoning instances whose hardness dynamically co-evolves with model capability under validation constraints.

## M    FUTURE WORKS

### M.1    GAME-THEORETIC FORMALIZATION OF THE ATAD PROTOCOL

A promising avenue for extending ATAD is to cast the Teacher–Orchestrator–Student loop itself as a cooperative game in which each agent's *move* (problem generation, validation, or solution) becomes a unit of experience whose marginal contribution to overall benchmark quality can be quantified with game-theoretic tools such as Shapley values and their ordered extensions — e.g., the Nowak-Radzik value that explicitly respects the temporal ordering of curriculum steps Diaz et al.. By treating successive rounds of ATAD as a sequence of coalitions, we could estimate how much each agent (or even each prompt strategy) accelerates difficulty calibration, then allocate compute or interaction budget proportionally to those cooperative gains; conversely, negative pairwise interactions would signal adversarial curricula or redundant checks that should be pruned. Embedding this credit-assignment mechanism inside the protocol would let ATAD adapt not only the problems it poses but also the roles and incentives of its constituent agents, yielding a self-tuning, game-theoretic benchmark that co-evolves with frontier LLMs while remaining transparent and fair. This transposition explicitly links ATAD's adaptive benchmarking to the proven game-theoretic curriculum framework Diaz et al., supplying both theoretical grounding and practical guidance for future protocol optimization.

### M.2    META-AGENT EXTENSIONS TO THE ATAD PROTOCOL

Recent advances in *meta agents*—agents that search over the design space of other agents—offer a promising path toward making ATAD self-improving, safer, and more sample-efficient. A meta

agent that *programs new agents in code*—as in Meta Agent Search Hu et al. (2025)—can iteratively discover superior teachers (richer problem generators) and orchestrators (sharper validators) while archiving each discovery for reuse. Complementarily, AFLOW's Monte-Carlo-Tree-Search over code-represented workflows can refine validation pipelines and student curricula so that even smaller models, paired with optimized workflows, rival larger baselines at a fraction of the compute cost Zhang et al. (2025). Casting *"make the problem just hard enough"*, *"catch adversarial trickery"*, and *"keep tasks unambiguous"* as explicit objectives in this unified search space lets the meta agent optimize difficulty, diversity, and alignment constraints simultaneously. Because each generated anomaly carries its provenance and (optionally) a machine-checkable proof, the resulting benchmark remains auditable even as it grows without bound. In short, a meta-agent layer transforms ATAD from a fixed three-role protocol into a self-refining ecosystem where agents, workflows, and evaluation criteria co-evolve alongside the LLMs they test.

## M.3 EXTENDING ATAD BEYOND INTERNAL-KNOWLEDGE REASONING

While the present work focuses on self-contained reasoning problems—aligned with standardized exam formats such as GRE, LSAT, and GMAT—the ATAD protocol is fundamentally task-agnostic and can be extended to incorporate external factual knowledge, mathematical axioms, or domain-specific constraints.

Below we outline concrete design directions enabled by the existing Teacher–Orchestrator–Student pipeline. These extensions were inspired in part by reviewer feedback, and we will explore them in future work.

### M.3.1 INTEGRATING EXTERNAL SOURCES INTO QUESTION GENERATION

The Teacher agent may be conditioned on retrieved factual snippets or curated external documents. For example:

- Factual reasoning: Provide the Teacher with retrieved Wikipedia or news snippets and instruct it to embed a domain-specific inconsistency.
- Scientific or legal reasoning: Supply the Teacher with short excerpts from scientific papers, laws, or technical specifications and ask it to introduce contradictions or violations of established rules.
- Mathematical reasoning: Supply explicit axioms, definitions, or numeric constraints and require the Teacher to generate perturbations consistent with a formal reasoning system.

These modifications only change the input to the Teacher; the Orchestrator's validation rules and Student-driven adaptive difficulty remain unchanged.

### M.3.2 RAG-ENHANCED ORCHESTRATOR VALIDATION

The Orchestrator can be augmented with retrieval components to verify the factual correctness or mathematical soundness of generated items. Examples:

- factual verification against retrieved knowledge bases
- unit-consistency or equation-validity checking
- symbolic-logic consistency checks for formal domains

Because the Orchestrator already performs strict structural and semantic validation, adding retrieval-augmented checks is conceptually straightforward.

### M.3.3 ADAPTIVE DIFFICULTY WITH EXTERNAL KNOWLEDGE

Difficulty escalation in ATAD currently intensifies semantic or logical challenge factors within self-contained contexts. In an external-knowledge setting, adaptive difficulty could escalate via:

- broader or less explicit supporting facts

- multi-hop reasoning over multiple retrieved snippets
- more abstracted or implicit references to factual background

This preserves the core ATAD principle: difficulty = increased reasoning demand, not increased ambiguity.

## M.4   ON GENERALITY BEYOND TEXT ANOMALY DETECTION

The focus on text anomaly detection in this work is driven by methodological considerations rather than domain restrictions. In contrast to math or code, where correctness criteria are strictly formalized and difficulty control is comparatively straightforward, text anomaly detection presents a uniquely challenging environment: difficulty must increase without introducing ambiguity, anomalies must remain subtle yet objectively identifiable, and reasoning must integrate pragmatics, discourse coherence, referential clarity, and multi-sentence inference.

This makes text anomaly detection a stringent *stress test* for dynamic difficulty scaling. Successfully controlling difficulty under such ambiguity-sensitive conditions provides a conservative validation setting for the ATAD protocol.

Importantly, ATAD is task-agnostic by design. Section 2 introduces no domain-specific assumptions; the protocol only requires a Teacher–Orchestrator–Student loop with validator-enforced clarity and correctness. As a result, the framework naturally generalizes to domains such as math-word inconsistencies, code-level reasoning, tool-use verification, and multimodal consistency.

Thus, rather than limiting generality, the present instantiation demonstrates ATAD's effectiveness in one of the most ambiguity-sensitive reasoning domains, providing strong evidence for transferability to more formally grounded tasks.

## M.5   ON REAL-WORLD REPRESENTATIVENESS AND FUTURE APPLICATIONS

In its current instantiation, ATAD adopts controlled GRE/LSAT-style anomalies to establish a clean diagnostic setting in which ground truth is precisely defined. This design choice enables controlled analysis of difficulty scaling under unambiguous reasoning conditions. Nevertheless, the ATAD protocol itself is not restricted to academic-style inputs.

The iterative, failure-driven difficulty escalation mechanism aligns closely with the types of reasoning breakdowns observed in real-world data, including subtle logical inconsistencies, causal misalignments, and procedural violations. Consequently, ATAD naturally extends to domain-grounded, real-world anomaly settings.

Representative future application domains include:

- **Finance**: causal inconsistencies or misaligned risk reasoning in analyst reports,
- **Scientific and medical writing**: methodological or interpretive inconsistencies,
- **Industrial operations**: violations of process logic, measurement coherence, or procedural constraints.

These domains involve exactly the types of reasoning-centric inconsistencies that the Teacher–Student competition and Orchestrator validation are designed to surface. Extending ATAD to such domain-grounded environments therefore constitutes a natural and practically meaningful next step.

## M.6   SUMMARY AND FUTURE DIRECTIONS

Together, the analyses in Sections M.4 and M.5 illustrate that ATAD's dynamic generation protocol is neither limited to text anomalies nor restricted to internal LLM knowledge alone. Instead, the protocol provides a flexible substrate for constructing domain-specialized dynamic benchmarks under validated difficulty control.

The successful application of ATAD to ambiguity-sensitive text anomaly detection demonstrates robustness under one of the most challenging reasoning environments. At the same time, the task-agnostic nature of the Teacher–Orchestrator–Student loop enables natural generalization to more

structured domains such as mathematical reasoning, code-level verification, tool-use consistency, and multimodal understanding.

Moreover, the failure-driven adaptive mechanism aligns closely with real-world reasoning breakdowns, suggesting strong potential for deployment in domain-grounded settings such as finance, scientific and medical analysis, and industrial operations.

Future work will therefore pursue (i) domain-specific instantiations of ATAD under external knowledge retrieval or conditioning, (ii) extensions to executable reasoning tasks such as code and tool use, and (iii) real-world benchmark construction in operational environments. These directions aim to further validate ATAD as a general-purpose protocol for dynamic, difficulty-regulated reasoning evaluation.

