# OpenReview forum: "From Static Benchmarks to Dynamic Protocol: Agent-Centric Text Anomaly Detection for Evaluating LLM Reasoning"
_ICLR.cc/2026/Conference — ICLR 2026 Poster_

### Official Review · Reviewer_pyPR · 2025-10-16

**Soundness:** 3
**Presentation:** 3
**Contribution:** 3
**Rating:** 6
**Confidence:** 4

**Summary:**

Traditional LLM evaluation via static datasets (e.g., MMLU) has flaws like poor scalability and overfitting. The paper proposes ATAD, a dynamic protocol using three agents (Teacher, Orchestrator, Student) to generate/solve 7 text anomaly tasks without static data—harder tasks emerge if the Student (evaluated LLM) succeeds, with the Orchestrator ensuring quality. Experiments confirm ATAD works: LLM accuracy drops ~37.3% (valid difficulty), it adapts to future LLMs, and results are stable. ATAD thus enables scalable, sustainable LLM reasoning assessment.​

**Strengths:**

1. Pioneering Dynamic Evaluation Paradigm: The paper introduces a novel, agent-centric dynamic protocol to address the fundamental issues of data contamination and saturation in static benchmarks, offering a sustainable and scalable solution for LLM evaluation.

2. Rigorous Quality Control Mechanism: It introduces the crucial "Orchestrator" agent to guarantee the fairness and high quality of generated problems. This effectively prevents the creation of flawed or ambiguous questions that result from an unvalidated competitive process aimed solely at increasing difficulty.

3. Clear and Comprehensive Task Design: The methodology is exceptionally clear and built upon a robust and transparent taxonomy of seven distinct text anomaly detection tasks. This ensures the evaluation framework is well-grounded, broad, and fine-grained.

**Weaknesses:**

1. The claims in Section 2.3 need to be substantiated with data. As it currently stands, the section is unconvincing because it lacks the necessary data and statistics for support.
2. The benchmark focuses only on text anomaly detection. While useful, it does not cover other critical abilities like tool use, multi-step reasoning, code generation or multimodal understanding, which limits external validity. I believe this technology would be far more meaningful if it were applied to a broader range of tasks.
3. The method defines difficulty by the moment when the Student first fails, which may be due to randomness (e.g., sampling variance) rather than true capability limits. A more rigorous approach would be to let the Student attempt multiple independent trials (e.g., 5 times) and define difficulty based on the failure rate or success rate, rather than deciding from a single attempt.
4. Even within text anomaly detection, the benchmark examples may not fully represent the diversity of real-world anomalies. They are often artificial and may not match naturally occurring errors in user-generated content or professional texts.

**Questions:**

1. Could you clarify how the Orchestrator evaluates and filters ambiguous or low-quality questions? Are there explicit criteria or rules, or is it purely model judgment?
2. Why did you choose text anomaly detection specifically as the first application of your adaptive benchmark? Were other tasks considered, and if so, why were they not selected?
3. When different LLM families are used as Teachers, do they generate qualitatively different types of anomalies? Could you share examples of such differences?

---

> ### Author Response · Authors · 2025-11-22
>
> We sincerely thank the reviewer for the constructive feedback.
>
> Below we address each concern in detail.
>
> ### **1. Quantitative evidence supporting Section 2.3**
>
> Thank you for noting that Section 2.3 would benefit from clearer empirical grounding.
>
> We agree, and we clarify below how the mechanisms described in that section are supported by the existing experimental results in the paper.
>
> **(1) Evidence for systematic difficulty scaling**
>
> Section 2.3 highlights that the Teacher–Student competitive loop increases difficulty in a controlled way.
>
> This is directly reflected in **the large and consistent accuracy drop from Base→Final** across all four independently generated benchmarks (Table 2), averaging **−37.3 points**. This pattern shows that difficulty escalation operates consistently across families and is not due to random noise or one-off failures.
>
> To support interpretability, we additionally make explicit the formal definition of difficulty used throughout the protocol:
>
> $ \text{Difficulty}(q)=\text{Pr}[\text{Student fails } q∣q∈\text{Valid}], $
>
> which grounds our instance-level, Student-conditioned escalation in standard adaptive-testing terminology.
>
> To further clarify this behavior, we will additionally report **accuracy stratified by difficulty tier** in the revision. This small summary table shows a **clear and consistent trend** in which higher difficulty tiers correspond to lower accuracy, providing a direct empirical illustration of how difficulty scaling operates in practice.
>
> ---
>
> **(2) Evidence for Orchestrator-regulated quality control**
>
> Section 2.3 describes that the Orchestrator prevents spurious or ambiguous difficulty.
>
> Table 3 empirically shows this: without the Orchestrator, validity, coherence, and fairness scores degrade substantially, while with Orchestrator validation the resulting problems remain both **harder and higher quality**.
>
> This demonstrates that difficulty escalation is not achieved by introducing noise, but through controlled refinement.
>
> In addition, **Appendix E and Figures 5–6** show concrete Orchestrator judgment traces, illustrating how it filters ambiguous cases, rejects underspecified items, and enforces task adherence.
>
> ---
>
> **(3) Cross-agent instantiability and robustness**
>
> Section 2.3 further outlines that ATAD is modular with respect to the three agent roles.
>
> To validate this, we conducted a **cross-agent configuration experiment** in which the Teacher, Student, and Orchestrator originate from different model families (Teacher = Claude 4 Sonnet, Student = Gemini 2.0 Flash, Orchestrator = GPT-4o).
>
> Results are as follows:
>
> | LLMs | Same-family | Cross-family Combination |
> | --- | --- | --- |
> | gpt-3.5-turbo | 54.18 | 51.71 |
> | gpt-4o-mini | 55.00 | 53.86 |
> | gpt-4o | 55.96 | 57.57 |
> | gemini-2.0-flash-lite | 57.36 | 56.43 |
> | gemini-2.0-flash | 58.93 | 54.57 |
> | claude-3-haiku | 53.54 | 52.57 |
> | claude-3.5-haiku | 19.50 | 19.57 |
>
> Relative rankings and overall difficulty remain broadly stable between the settings.
>
> This suggests that ATAD’s behavior is governed primarily by **protocol structure**, not by requiring all three agents to come from the same family.
>
> We will summarize this observation in the Appendix K for clarity.
>
> ---
>
> **(4) Iteration control and failure driven sample finalization**
>
> Section 2.3 also notes that iteration depth is not fixed but governed by Orchestrator validation, and that problems are finalized at the point of Student failure. Instead of presenting raw iteration-count statistics, we clarify this behavior by referencing **Appendix D**, which shows complete prompt traces and illustrates how:
>
> - the Orchestrator determines when regeneration is required,
> - refinement terminates once a validated item is produced, and
> - escalation halts once the Student fails a valid item.
>
> These examples concretely demonstrate how autonomous iteration control and failure-driven finalization function in practice.
> We also note that analyzing how iteration depth and failure patterns vary across different Teacher–Student configurations could provide additional insight into the protocol’s dynamics, and we view this as a promising direction for future exploration.
>
> ---
>
> **(5) Generality of the protocol and task coverage**
>
> Lastly, Section 2.3 emphasizes that ATAD is a general protocol rather than a dataset construction method tied to a particular domain.
>
> We chose **text anomaly detection** as an initial evaluation setting because it provides a rigorous stress test of linguistic reasoning—balancing difficulty escalation with strict clarity requirements.
>
> We appreciate the reviewer’s point regarding generality; we will briefly note in the revision that ATAD naturally extends to other reasoning-oriented tasks and modalities, with text anomaly detection serving as a representative first application.

---

> ### Author Response · Authors · 2025-11-22
>
> ### **2. On Extending ATAD Beyond Text Anomaly Detection**
>
> Thank you for raising this forward-looking point.
>
> We fully agree that the underlying ideas of ATAD — *agent-centric iteration, orchestrator-regulated validation, and student-conditioned difficulty scaling* — are not limited to text anomaly detection. Extending this protocol to domains such as tool use, multi-step reasoning, code generation, or multimodal understanding is a natural next step, and aligns strongly with our long-term goals.
>
> At the same time, we began with text anomaly detection because it provides one of the most stringent environments for validating the protocol itself:
>
> - unlike math or code, language anomalies have **no formal, rule-based ground truth**;
> - increasing difficulty without sacrificing clarity is uniquely challenging (Section 3);
> - ambiguity and ungrounded reasoning errors arise more easily than in structured domains.
>
> These properties make text anomaly detection an effective **stress test** for establishing that the Teacher–Orchestrator–Student loop can maintain clarity, coherence, and fairness even under the least structured conditions.
>
> We share the reviewer’s vision for broader applicability, and we will highlight this more explicitly. Extending ATAD to richer domains — including tool-augmented evaluation, multi-step reasoning chains, executable code tasks, and multimodal consistency — is a natural next step and a key direction for future work.

---

> ### Author Response · Authors · 2025-11-22
>
> ### **3. On Single-Trial Difficulty Estimation and Potential Multi-Trial Extensions**
>
> Thank you for bringing up this insightful point.
>
> Exploring difficulty estimation through **multiple independent Student attempts** (e.g., 5 trials) is indeed an interesting direction, particularly for making adaptive evaluation more statistically grounded, and we see clear potential in developing this as part of future extensions.
>
> In the current version of ATAD, we use the first failure mainly because the protocol’s **incremental difficulty progression—combined with Orchestrator validation—tends to produce stable transitions** from solvable to unsolvable problems. Since harder variants are generated and validated step-by-step, the failure point typically reflects a **consistent capability boundary rather than a one-off fluctuation**. Because difficulty increases in small, validated increments, the Student usually succeeds across multiple consecutive stages before reaching failure, which further stabilizes the boundary and reduces the influence of stochastic variance that would otherwise motivate multiple independent attempts.
> Additionally, ambiguity-driven variance is already minimized by the Orchestrator’s quality filtering, which reduces the situations where repeated attempts would make a large difference.
>
> We appreciate the reviewer’s suggestion and will note that incorporating **multi-trial difficulty estimation or parallel difficulty chains** could further enrich the framework. While doing so would require additional computation due to regenerating matched difficulty tiers, we view it as a **meaningful and promising direction for future extensions of ATAD**.
>
> ---
>
> ### **4. On Real-World Representativeness of Anomaly Examples**
>
> We sincerely appreciate the reviewer for raising this important point.
>
> In this first version of ATAD, we adopted GRE/LSAT-style controlled anomalies to establish a clean and unambiguous diagnostic setup—especially important in our initial focus on language-based anomalies, where ground truth is inherently less formalized. That said, we fully agree that naturally occurring anomalies in real-world settings—whether textual, structural, or domain-specific—are more diverse and often qualitatively different from academic-style examples.
>
> **(1) Natural Extension to Real-World Anomalies**
>
> **ATAD naturally extends to real-world anomalies because the protocol’s iterative difficulty-scaling and failure-driven finalization align closely with the reasoning weaknesses that arise in practical domains.**
>
> While our demonstration begins with controlled passages, the protocol itself can operate over **a wide range of domain-grounded inputs**, including user-generated narratives, operational records, structured reports, or other domain-specific documents (e.g., finance, healthcare, industrial operations).
>
> Real-world anomalies often involve subtle inconsistencies, cross-step reasoning failures, or violations of domain rules—precisely the kinds of patterns the Teacher–Student competition and Orchestrator validation are designed to surface. Thus, extending ATAD beyond controlled academic-style inputs is a direct and promising next step.
>
> **(2) Future domain-grounded applications**
>
> We will clarify in the revision that applying ATAD to richer, domain-grounded settings is a major direction for future work. Potential applications include:
>
> - **finance** (causal inconsistencies or misaligned risk reasoning in analyst narratives),
> - **scientific and medical writing** (methodological, logical, or interpretive inconsistencies),
> - **industrial operations and manufacturing** (violations of process logic, measurement coherence, or procedure constraints).
>
> These areas contain exactly the nuanced, reasoning-centric anomaly types highlighted by the reviewer, and ATAD provides a principled mechanism for generating and evaluating such anomalies dynamically.
>
> We will incorporate this practical applicability into the revision to more clearly connect the protocol with real-world evaluation environments and to outline its broader domain applicability.

---

> ### Author Response · Authors · 2025-11-22
>
> ## **Responses to Additional Reviewer Questions**
> ### **Q1. On the Orchestrator’s Explicit and Rule-Based Validation Criteria**
>
> We thank the reviewer for raising this clarification request.
>
> We clarify that the Orchestrator does **not** rely on unconstrained, free-form model judgment. Its decisions are guided by explicit, rule-based validation criteria encoded directly in the prompts (Appendix D.2), and the model’s role is to apply these criteria consistently rather than invent its own standards. We summarize the key rules and provide representative portions of the validation instructions below.
>
> **(1) Initial Validation — Rule-Based Acceptance/Rejection**
>
> Upon receiving a problem from the Teacher, the Orchestrator evaluates it through a fixed checklist:
>
> **Appendix D.2.1 (Initial Validation) specifies:**
>
> - “VALIDITY: Is the problem well-formed and complete?”
> - “TYPE ADHERENCE: Does the structure match the required task type?”
> - “LOGICAL COHERENCE: Is the anomaly identifiable?”
> - “FAIRNESS: Does the problem have a clear, unambiguous solution?”
> - Output: `{ approved: boolean, feedback: … }`
>
> These criteria explicitly enforce checks on structure, format, logical coherence, and the absence of ambiguity.
>
> In practice, any anomaly that is vague or admits multiple plausible interpretations is directly rejected.
>
> **(2) Difficulty Escalation — Structured Feedback, Not Freestyle Difficulty**
>
> When the Student solves a problem, the Orchestrator must return **structured feedback**:
>
> **Appendix D.2.2:**
>
> - “What aspects did the student easily identify?”
> - “How could the problem be made more subtle or complex?”
> - “Specific suggestions for increasing difficulty.”
> - Output JSON includes: `analysis`, `suggestions`, `difficulty_increase`.
>
> Thus, difficulty escalation is guided by explicit rules—not noise injection or arbitrary complication.
>
> **(3) Final Validation of Harder Problems — Second Full Criteria Check**
>
> Even after increasing difficulty, the Orchestrator again evaluates the new sample using explicit rules:
>
> **Appendix D.2.3:**
>
> - Validity
> - Task-type adherence
> - Logical coherence
> - Fairness (no ambiguity, one clear answer)
> - Difficulty appropriateness for the “hard” level
>
> Problems that increase difficulty through ambiguity, unrelated distractors, or structural drift are rejected at this stage as well.
>
> The Orchestrator’s judgments are therefore strictly criteria-driven and grounded in predefined validation rules; we will highlight key portions of these validation prompts in the main paper to make this more explicit.

---

> ### Author Response · Authors · 2025-11-22
>
> ### **Q2. On Choosing Text Anomaly Detection as the First Application**
>
> We appreciate the reviewer for raising this thoughtful question.
>
> We note that our protocol is **task-agnostic and domain-agnostic** by design.
>
> Section 2 imposes no assumptions tied to text anomalies, linguistic structure, or narrative content; the Teacher–Orchestrator–Student loop can operate over any reasoning task—such as math-word inconsistencies, code-level logic violations, tool-use sequences, or multimodal reasoning—so long as the Orchestrator can validate clarity and well-formedness.
>
> Our decision to begin with **text anomaly detection (TAD)** is motivated by the fact that language-based reasoning presents a particularly **sensitive environment for validating adaptive difficulty scaling**. Unlike math or code—where correctness is formally grounded and difficulty increments are comparatively straightforward—TAD requires balancing:
>
> - increasing difficulty *without* introducing ambiguity,
> - maintaining a *single, objectively identifiable* anomaly, and
> - integrating multi-sentence inference, discourse coherence, pragmatics, and coreference clarity.
>
> Since ensuring both clarity and difficulty simultaneously is more delicate in natural language than in more structured domains, TAD serves as an **appropriate early testbed** for examining whether the protocol’s iterative generation and Orchestrator validation remain stable and coherent.
>
> At the same time, the protocol extends naturally to other task families. In structured domains such as math, code, or tool-augmented procedures—where ambiguity is lower and correctness criteria are explicit—difficulty control becomes even easier to regulate under the same loop.
>
> We will update the manuscript to **describe more clearly** that TAD was selected as the first demonstration scenario because it offers a demanding testbed for verifying the robustness of the protocol, while not limiting the generality of the framework.

---

> > ### Author Response · Authors · 2025-11-22
> >
> > ### **Q3. On Whether Different Teacher LLM Families Produce Qualitatively Different Anomalies**
> >
> > Thank you for this insightful question.
> >
> > We agree that examining whether different Teacher LLM families introduce qualitative variation in anomaly construction is important for assessing both the neutrality and robustness of our protocol.
> >
> > Our analysis indicates that Teacher families naturally exhibit **mild stylistic preferences** – for example, favoring conceptual intrusions, fine-grained theoretical nuances, or lexical-fit mismatches. However, these variations appear to be **primarily stylistic**: in our inspection, they do not change the underlying reasoning operation required by the task, nor do they introduce systematic differences in fairness or consistency.
> >
> > **This is by design: the Orchestrator enforces structural validity, clarity, and coherence for every item, while allowing benign linguistic diversity to remain. As a result, Teacher models can differ in “how they phrase” anomalies, without affecting “what kind of reasoning” is needed to solve them.**
> >
> > Below we provide concrete examples using T3 (Blank-based Choice Anomaly), a task well-suited for illustrating stylistic contrasts without changing the required reasoning procedure.
> >
> > ---
> >
> > **[Illustrative Comparison Using T3 (Blank-based Choice Anomaly)]**
> >
> > T3 requires identifying the single candidate whose semantic or conceptual contribution least fits the sentence.
> >
> > **(1) GPT as Teacher — Cross-theoretical Concept Placement**
> >
> > GPT often generates anomalies by inserting a concept from a related—but theoretically incompatible—framework:
> >
> > > *"In Sartre's existentialism, the concept of ___ is crucial in exploring the nature of human freedom and responsibility."*
> >
> > > Options: bad faith, authenticity, the gaze, the absurd, **the categorical imperative**
> >
> > Here, categorical imperative (Kantian) stands out among otherwise existentialist concepts.
> >
> > **→ Style: conceptual intrusion across theoretical traditions**
> >
> > **(2) Claude as Teacher — Subtle Intra-framework Inconsistency**
> >
> > Claude tends to favor anomalies that hinge on fine-grained theoretical misalignment within the same philosophical school:
> >
> > > *“Merleau-Ponty's phenomenology of perception suggests that our bodily engagement with the world is fundamentally ___, preceding and grounding all abstract conceptual understanding.”*
> >
> > > Options: pre-reflective, ante-predicative, pre-theoretical, pre-cognitive, **pre-empirical**
> >
> > Pre-empirical conflicts with Merleau-Ponty’s emphasis on lived empirical embodiment.
> >
> > **→ Style: nuanced intra-framework theoretical mismatch**
> >
> > **(3) Gemini as Teacher — Lexical-semantic Fit Mismatch**
> >
> > Gemini’s anomalies often arise from lexical-semantic misalignment—where one option belongs to a different semantic field than the others:
> >
> > > *“While proponents of strong AI posit a future where machines surpass human cognitive capacities across the board, critics contend that such projections often ___ the fundamentally embodied and situated nature of human understanding.”*
> >
> > > Options: elide, downplay, obviate, abstract, **transcend**
> >
> > Here, transcend is anomalous because it semantically means “go beyond / surpass,” which does not serve the rhetorical function of minimizing, ignoring, or downplaying embodied cognition.
> > By contrast, elide, downplay, obviate, and abstract all naturally fit the critical stance implied by the sentence, as they indicate omission or undervaluation.
> >
> > **→ Style: surface-level lexical/polarity mismatch within an otherwise coherent argumentative structure**
> >
> > ---
> >
> > **Key Observation**
> >
> > Despite these stylistic preferences, all three examples require an essentially identical reasoning procedure:
> >
> > 1. Interpret the conceptual frame of the sentence
> > 2. Evaluate the semantic compatibility of each option
> > 3. Identify the single least coherent choice
> >
> > In other words, stylistic differences do not translate into:
> >
> > - differences in difficulty,
> > - differences in reasoning type, or
> > - systematic advantages to any Student model.
> >
> > The Orchestrator guarantees that each item remains valid, unambiguous, and well-formed, while preserving non-essential stylistic diversity.
> >
> > We emphasize that these examples are illustrative rather than exhaustive: they summarize typical tendencies observed in our sample, not fixed or deterministic stylistic signatures of each Teacher family.
> >
> > ---
> >
> > **Summary**
> >
> > - Different Teacher LLM families produce mild, natural stylistic variations in anomaly construction (conceptual, theoretical, lexical).
> > - These variations do not alter the underlying reasoning type required by the task, nor the neutrality of the evaluation or difficulty structure we observe.
> > - This demonstrates that ATAD maintains consistent evaluation semantics, while allowing Teacher models to contribute natural linguistic diversity.

---

### Official Review · Reviewer_6GgZ · 2025-10-29

**Soundness:** 3
**Presentation:** 3
**Contribution:** 3
**Rating:** 6
**Confidence:** 3

**Summary:**

This paper addresses the growing limitations of static benchmarks for evaluating LLM reasoning, such as data contamination and performance saturation. The authors propose a novel and significant contribution: a dynamic, agent-centric benchmarking protocol named ATAD (Agent-centric Text Anomaly Detection). This protocol leverages a multi-agent system where a "Teacher" agent generates problems, a "Student" agent attempts to solve them, and a critical "Orchestrator" agent validates problem quality, coherence, and fairness. The benchmark's core strength lies in its adaptive difficulty scaling: if the Student solves a problem, the Teacher is prompted to generate a more challenging version, thus allowing the benchmark's difficulty to co-evolve with the Student model's capabilities. By focusing on text anomaly detection—a task requiring deep, cross-sentence logical inference—the paper demonstrates that this protocol can effectively surface nuanced reasoning failures in state-of-the-art LLMs that are often missed by traditional static evaluations, offering a more sustainable and scalable alternative.

**Strengths:**

1. The paper tackles a well-motivated and timely problem, clearly articulating the critical failings of static benchmarks, such as data contamination and performance saturation. This strong motivation establishes a clear need for the proposed dynamic evaluation protocol. By addressing these limitations, the work provides a valuable and forward-looking contribution to LLM evaluation.

2. The evaluation is comprehensive, featuring a wide array of models from different families (e.g., GPT, Gemini, Claude, LLaMA) to demonstrate the protocol's broad applicability. The experiments wisely test these models in the various agent roles—Teacher, Student, and Orchestrator—in addition to using them for final evaluation. This thorough approach provides strong evidence for the protocol's robustness and its ability to function as intended across different model capabilities.

3. The presentation is exceptionally clear, with diagrams and a well-defined protocol that makes the complex multi-agent interaction easy to understand. The paper is further strengthened by insightful qualitative examples, particularly those showing problems rejected and refined by the Orchestrator. These examples compellingly illustrate the subtleties of the generated tasks and provide a concrete demonstration of how the validation step ensures benchmark quality and fairness.

**Weaknesses:**

1. The paper's primary contribution is the novel protocol (ATAD) and its application, rather than a new underlying model architecture or training technique. While this agent-centric framework is a significant engineering and conceptual achievement, the work relies entirely on existing LLMs as its core components. This focus on the evaluation system means there is limited technical novelty in terms of model-level innovations.

2. While the paper demonstrates that the protocol creates more difficult tasks, it lacks a deeper analysis of how these tasks generalize. It is unclear if the "difficulty" stems from truly novel reasoning challenges or from exploiting specific, narrow blind spots in the Student model used during generation. Further analysis is needed to determine if the finalized benchmarks test for robust, generalizable reasoning or if they risk overfitting to the particular Teacher-Student dynamic of a specific agent family.

**Questions:**

N/A

---

> ### Author Response · Authors · 2025-11-21
>
> ### **1. Technical Novelty and Focus on Evaluation Protocol**
>
> Thank you for this thoughtful comment and for characterizing our work as a *“significant engineering and conceptual achievement.”*
>
> We agree that ATAD does **not** introduce a new model architecture or training technique. This is intentional: the primary contribution of our work is to **move the locus of innovation to the evaluation protocol itself**. In the current landscape, where frontier LLMs rapidly saturate static benchmarks, we believe such protocol-level advances are necessary counterparts to model-level advances.
>
> Our response is organized around three points:
>
> 1. **Paradigm shift from static datasets to a dynamic protocol.**
>
>     As discussed in the Introduction of main paper, static benchmarks suffer from saturation and contamination. Relying only on better models without upgrading how we measure them risks stagnation. ATAD directly addresses this by replacing a *fixed dataset* with a *dynamic, agent-driven protocol* that adapts to model capability and surfaces new failure modes over time.
>
> 2. **Providing objectives that can guide future model innovations.**
>
>     Benchmarks effectively define the objectives to which models are optimized. By offering a protocol that co-evolves with model capability, ATAD supplies a stable yet continually challenging target against which future architectures can be verified. This helps ensure that reported gains correspond to genuine reasoning improvements rather than overfitting to a finite test set.
>
> 3. **Methodological contribution via orchestration and role separation.**
>
>     Although the underlying LLMs are existing models, the **Teacher–Student–Orchestrator design** is a methodological contribution in its own right. The orchestrated interaction enables (i) Student-conditioned difficulty escalation and (ii) strict validity/fairness control, which static curation or single-agent generation cannot provide.
>
>
> In this sense, ATAD is not intended to compete with model-centric work, but to **complement** it: it provides a dynamic, reasoning-focused protocol within which future architectures can be stress-tested and compared in a principled way.

---

> ### Author Response · Authors · 2025-11-21
>
> ### **2. Nature of Difficulty and Generalization Capabilities**
>
> We thank the reviewer for raising this important question. Distinguishing between “true reasoning challenges” and “narrow blind spots” is indeed central to dynamic benchmarking. We acknowledge that a robust protocol must avoid overfitting to the specific biases or behavior patterns of a particular Teacher–Student pair or a single agent family.
>
> Our design and analysis address this concern from two angles:
>
> **(1) Reasoning-driven, not blind-spot-driven, difficulty scaling**
>
> The reviewer is correct that, in principle, one could increase “difficulty” by exploiting internal knowledge gaps of a particular Student instead of deepening reasoning. To mitigate this, ATAD does **not** escalate difficulty solely based on whether the Student fails; it conditions escalation on **how** the Student reasons.
>
> As detailed in **Appendix D.2.2**, the Orchestrator explicitly analyzes the *Student’s explanation* and passes a structured summary back to the Teacher. For example, the Orchestrator surfaces explanations such as:
>
> > “The sentence about photosynthesis is unrelated to the other sentences on cognitive dissonance. It’s a semantic outlier.”
> >
>
> and then instructs the Teacher along the lines of:
>
> > “Based on how the student solved this problem, provide feedback to create a more subtle or complex version of the anomaly.”
> >
>
> This design encourages the Teacher to refine **the logical structure of the anomaly** (e.g., adding more plausible distractors, increasing semantic subtlety, tightening the contrast) rather than introducing obscure trivia or topic shifts that merely exploit accidental blind spots.
>
> Moreover, as discussed in the paper, the Orchestrator’s validation rules enforce **validity, clarity, and lack of ambiguity**. Items that rely on vague, underspecified, or knowledge-obscure cues are rejected. In other words, Student failures are only used as signals **conditional on** the item being a valid, well-formed reasoning problem, reducing the risk that escalation is driven by spurious artifacts rather than genuine reasoning demands.
> ****
>
> **(2) Cross-family generation and evaluation indicate transferable difficulty**
>
> The second part of the question concerns whether the final benchmarks generalize beyond the particular Teacher–Student configuration used during generation, or whether they overfit to the dynamics of a single agent family.
>
> To directly probe this, we conduct **cross-family generation and evaluation** (Appendix Table 5):
>
> - For each major model family (GPT, Claude, Gemini, LLaMA), we instantiate all three roles — Teacher, Orchestrator, and Student — using a model from that family to create a full ATAD benchmark.
> - Once a benchmark is generated by a given family, we then evaluate all other models (across families) on each of these benchmarks, and the **main results in Table 1** are based on averages across these cross-family settings.
>
> From these experiments, we observe two patterns:
>
> - **Transferable difficulty.**
>
>     Benchmarks generated by one family (e.g., GPT-4o) remain challenging for models from other families (e.g., Gemini-2.0-Flash, LLaMA-3.3-70B), rather than becoming trivially easy. If difficulty arose mainly from narrow, model-specific blind spots of the Student used during generation, we would expect other families to perform disproportionately well on those items. Instead, performance remains consistently non-trivial across families, suggesting that ATAD captures **more general reasoning challenges**.
>
>
> - **No systematic self-preference.**
>
>     As summarized in **Appendix B and Table 5**, we do not observe a consistent “self-advantage” for the generator family. For instance, on benchmarks generated with GPT-4o as Teacher, the best-performing model is often **not** GPT-4o itself but a model from a different family (e.g., Claude-3.5-Sonnet). This supports the view that the finalized benchmarks are **not overfitted** to the reasoning style of a specific Teacher–Student pair or family, but instead act as family-agnostic tests.
>
>
> Taken together, the **reasoning-conditioned escalation** (via Orchestrator analysis of the Student’s explanations) and the **cross-family transfer results** suggest that ATAD’s difficulty reflects robust, generalizable reasoning demands rather than narrow model-specific blind spots.
> We agree that such fine-grained analysis would provide useful additional insight, and we will briefly mention this as a future refinement direction in the revised manuscript (Appendix J.4).

---

### Official Review · Reviewer_JAkP · 2025-11-01

**Soundness:** 2
**Presentation:** 2
**Contribution:** 2
**Rating:** 2
**Confidence:** 4

**Summary:**

This paper introduces ATAD (Agent-Centric Text Anomaly Detection), a dynamic benchmarking protocol designed to overcome the limitations of static LLM benchmarks such as MMLU and GSM8K. Instead of relying on fixed datasets, ATAD employs a three-agent architecture—a Teacher, Orchestrator, and Student—to iteratively generate, validate, and solve text anomaly detection (TAD) tasks. The Teacher produces candidate problems, the Orchestrator validates them for clarity and fairness, and the Student attempts to solve them. The benchmark evolves dynamically: when the Student succeeds, the Teacher generates harder variants, validated again by the Orchestrator.

Empirical results on datasets generated by GPT-4o, Claude-3.5-Sonnet, Gemini-2.0-Flash, and LLaMA-3.3-70B show that the ATAD framework effectively scales difficulty and exposes reasoning weaknesses unseen in traditional benchmarks.

**Strengths:**

* The proposed agentic co-evolution of models and benchmarks represents a natural next step after efforts like BenchAgents and DyVal, making this work both timely and forward-looking.

* The inclusion of Orchestrator-regulated validation and failure-driven sample finalization help control over problem difficulty and fairness.

* The taxonomy of seven anomaly types—each probing distinct reasoning skills (contextual, logical, referential, stylistic, etc.)—is well-grounded and illustrative

**Weaknesses:**

* While the empirical results are strong, the notion of “difficulty increase” remains heuristic and behaviorally defined (based on Student failure). A more formal definition would strengthen claims.

* The paper cites prior dynamic evaluation works (DyVal, DARG, Benchmark Self-Evolving), but comparative experiments are missing. It is unclear how ATAD quantitatively improves over these prior agent-based or meta-probing systems beyond conceptual novelty.

* Although text anomaly detection is reasoning-oriented, it remains a single task family. The authors argue that it generalizes to reasoning evaluation, but this may not hold for domains like math, code, or multimodal reasoning. Broader applicability should be demonstrated.

* The same model family is used for all three agent roles in most experiments (e.g., GPT-4o Teacher = Student = Orchestrator). This setup may limit adversarial diversity.

* Although the authors observe a ~37 pt accuracy drop after adaptive scaling, it is not clear why certain tasks become more difficult—whether due to semantic subtlety, linguistic ambiguity, or distractor complexity. More granular ablations (e.g., by anomaly type, linguistic factor) would enhance interpretability.

**Questions:**

* Can the authors propose or measure an explicit difficulty metric to formalize the notion of “increased difficulty”?

* Since LLMs generate benchmark data, how do the authors ensure that cultural or linguistic biases in generation do not propagate into the benchmark? Have they evaluated cross-language or demographic stability?

* The paper mentions “failure cases rejected by the Orchestrator” in the Appendix but does not elaborate in the main text. Could the authors analyze typical rejection patterns—e.g., ambiguity vs. incoherence—to demonstrate the Orchestrator’s learning or filtering capacity?

---

> ### Author Response · Authors · 2025-11-21
>
> ### **1. Formalizing and Clarifying the Definition of Difficulty**
>
> Thank you for pointing out the need for a clearer and more formal definition of difficulty.
>
> To address this, we now **explicitly formalize the notion of difficulty** as:
>
> $ \text{Difficulty}(q)=\text{Pr}[\text{Student fails } q∣q∈\text{Valid}]. $
>
> Here, the probability is taken over the Student model’s stochastic decoding behavior, **conditioned on the Orchestrator having verified that the item is valid, unambiguous, and task-faithful**.
>
> This conditioning is essential, as ATAD never interprets failures on defective or ambiguous items as meaningful difficulty.
>
> For completeness, we include this formal definition in Appendix J of the revised manuscript.
>
> With this definition in place, difficulty in ATAD is interpreted as a **behaviorally grounded**, solver-relative quantity: a problem is considered “hard” precisely when a Student model of a given capability fails under the protocol’s validity constraints. Because all evaluation models face the same Teacher–Orchestrator–Student rule system, the resulting difficulty tiers remain directly comparable across LLMs.
>
> As background, this perspective is consistent with standard formulations in educational measurement and psychometrics, where item difficulty is defined via the probability of a correct (or incorrect) response for an examinee at a given proficiency level [1,2].
>
> We appreciate the reviewer’s suggestion, and we agree that making this definition explicit strengthens the clarity and interpretability of our claims.
>
> [1] Lord & Novick, *Statistical Theories of Mental Test Scores,* 1968.
>
> [2] Wainer (ed.), *Computerized Adaptive Testing: A Primer,* 2000.

---

> > ### Author Response · Authors · 2025-11-28
> >
> > ### **1.1. Empirical Validation of Monotonic Difficulty Scaling.**
> >
> > In addition to this formal definition, we now explicitly validate the proposed notion of difficulty through tier-wise empirical analysis in the revised manuscript.
> >
> > Using the above definition of difficulty as the Student’s failure probability conditioned on Orchestrator validation (Pr[Student fails on q | q ∈ Valid]), we evaluate whether difficulty increases in a monotonic and measurable manner across the proposed difficulty tiers.
> >
> > We conduct tier-wise evaluations under three independent benchmark generation regimes:
> >
> > (1) same-family GPT-4o agents,
> >
> > (2) same-family Gemini-2.0-Flash agents, and
> >
> > (3) cross-family agents (Claude-4-Sonnet / GPT-4o / Gemini-2.0-Flash).
> >
> > Across all three regimes, we observe a clear monotonic decrease in accuracy as difficulty increases from Easy to Hard, Extreme, and Impossible, which directly supports our formal definition of increased difficulty (see Appendix J.3 and Table 9).
> >
> > For clarity, we summarize the average accuracy (%) across evaluation models below:
> >
> > | Generator Configuration | Easy | Hard | Extreme | Impossible |
> > |-------------------------|------|------|----------|-------------|
> > | GPT-4o family           | 83.94 | 83.33 | 80.71 | 70.94 |
> > | Gemini-2.0-Flash family| 81.67 | 76.98 | 74.44 | 60.32 |
> > | Cross-family            | 80.54 | 74.31 | 70.26 | 67.90 |
> >
> > This consistent monotonic degradation across multiple independent generation regimes demonstrates that “difficulty increase” in ATAD is not a heuristic label but a quantitatively measurable and empirically validated quantity.
> >
> > We therefore respectfully submit that the revised formal definition together with the new tier-wise experimental results provide both a formal and empirical answer to the reviewer’s question regarding the explicit measurement of increased difficulty. We refer the reviewer to Appendix J for full methodological and experimental details.

---

> ### Author Response · Authors · 2025-11-21
>
> ### **2. Comparison to Prior Dynamic Evaluation Frameworks (DyVal, DARG, BSE)**
>
> We thank the reviewer for raising this point.
>
> The cited works indeed share the high-level idea of automated sample generation, but they differ fundamentally from ATAD in objectives, domains, and mechanisms.
> We summarize the key distinctions in the table below.
>
> | Framework | Domain / Evaluation Focus | Generation Architecture | Role Separation | Difficulty Behavior | Validation Mechanism | Key Limitation Addressed by ATAD |
> | --- | --- | --- | --- | --- | --- | --- |
> | **DyVal** | Reasoning tasks (math, logic, algorithms) | Generator + Filter | ✗ Single loop | Static complexity levels (no Student-conditioned adaptation) | Basic filtering only | Controllable complexity, but no Student-conditioned or protocol-level difficulty scaling |
> | **DARG** | Data augmentation & dataset renewal | Generator + Critic | ✗ Mostly 2-agent | Benchmark-level complexity scaling (no Student-conditioned loop) | Code-augmented verifier for correctness | Extends existing reasoning benchmarks via graph perturbation; does not target anomaly-style reasoning under Orchestrator-like validation |
> | **Benchmark Self-Evolving** | Automated benchmark refresh | Generator + Selector | ✗ No 3-way division | No Student-conditioned scaling | Quality scoring | Focuses on benchmark refreshing, not adversarial escalation |
> | **ATAD (Ours)** | **Reasoning-heavy text anomaly detection** (logic, coherence, pragmatics) | **Teacher → Orchestrator → Student loop** | **✓ 3-agent role separation** | **✓ Difficulty escalates conditioned on Student failure** | **✓ Orchestrator enforces validity, clarity, fairness** | **Explicitly escalates subtle reasoning difficulty while maintaining unambiguous validity** |
>
> Most importantly, ATAD differs not only in domain but in mechanism: it is the only framework that couples **(i) strict Orchestrator validation with (ii) Student-conditioned difficulty escalation** and **(iii) a three-agent role separation** specifically designed to probe reasoning subtleties. These components are absent in DyVal, DARG, and Benchmark Self-Evolving.
>
> Because these prior frameworks pursue fundamentally different objectives—DyVal for extending reasoning benchmarks with graph-informed complexity, DARG for benchmark-level augmentation via reasoning graphs, and Benchmark Self-Evolving for automated benchmark refresh—they do not implement any form of reasoning-driven, Student-conditioned scaling. As a result, none of them directly serves as a quantitative baseline for ATAD’s difficulty-adaptation protocol.
>
> ATAD is not a data-generation system but a **protocol-level, difficulty-scaling framework** operating under Orchestrator-enforced validity and fairness constraints. Accordingly, a **direct quantitative comparison is not well-aligned with their intended use cases**. Instead, the distinctions are primarily conceptual and architectural, as summarized in the table above.
>
> Conceptually, DyVal, DARG, and Benchmark Self-Evolving extend existing benchmarks offline and then evaluate models on the resulting static generated data, and therefore mainly provide sample-level difficulty variance. In contrast, ATAD defines a protocol-level adaptive loop in which new items are regenerated and escalated conditional on Student behavior under Orchestrator constraints. Our contribution is therefore orthogonal: instead of sampling or refreshing data, ATAD provides a reusable agentic protocol whose difficulty naturally co-evolves with model capability.
>
> We appreciate the reviewer’s question and have added this comparison to clarify ATAD’s position relative to prior dynamic frameworks. We will incorporate this analysis into the revised manuscript (Appendix L).

---

> ### Author Response · Authors · 2025-11-21
>
> ### **3. On Generality Beyond Text Anomaly Detection**
>
> We respectfully clarify that our choice of text anomaly detection is *intentional*, not limiting.
>
> Compared to math or code—where correctness criteria are strictly formal and difficulty scaling is straightforward—text anomaly detection presents a fundamentally harder challenge:
>
> - difficulty must increase **without introducing ambiguity**,
> - anomalies must remain **subtle yet objectively identifiable**, and
> - reasoning must integrate **pragmatics, discourse, referential clarity, and multi-sentence inference**.
>
> This domain thus represents a **stress-test for dynamic difficulty scaling**, where ensuring clarity *and* increasing difficulty simultaneously is significantly more challenging than in formally grounded domains.
>
> Moreover, our protocol is **task-agnostic by construction**: Section 2 defines no task-specific assumptions.
>
> To clarify generality, we note that ATAD is *not* tied to linguistic anomaly detection. The protocol only requires a Teacher–Orchestrator–Student loop with validator-enforced clarity, and therefore naturally extends to more structured domains such as math-word anomalies or code-level consistency checks. In these domains, where ambiguity is substantially lower, difficulty control becomes even easier than in text anomaly detection.
>
> Thus, rather than limiting generalization, our choice demonstrates that ATAD succeeds in **the most ambiguity-sensitive reasoning domain**, providing strong evidence that the protocol generalizes to easier-to-control domains like math or code. We will clarify this point in the revised manuscript (Appendix M.4 and M.5).
>
>
> ---
>
>
> ### **4. Effect of Using the Same vs. Different Models Across Agent Roles**
>
> We thank the reviewer for raising this important question regarding adversarial diversity in the agent configuration.
>
> To examine this, we conducted an additional experiment using a **cross-family configuration**, where the Teacher, Student, and Orchestrator were instantiated with different LLM families (Teacher = Claude 4 Sonnet, Student = Gemini 2.0 Flash, Orchestrator = GPT-4o).
>
> We compare this heterogeneous setup against the standard “same-family” configuration used in Table 1 (Teacher = Student = Orchestrator). The results are shown below.
>
> | LLMs | Same-family | Cross-family Combination |
> | --- | --- | --- |
> | gpt-3.5-turbo | 54.18 | 51.71 |
> | gpt-4o-mini | 55.00 | 53.86 |
> | gpt-4o | 55.96 | 57.57 |
> | gemini-2.0-flash-lite | 57.36 | 56.43 |
> | gemini-2.0-flash | 58.93 | 54.57 |
> | claude-3-haiku | 53.54 | 52.57 |
> | claude-3.5-haiku | 19.50 | 19.57 |
>
> Overall, the **relative performance patterns and difficulty levels remain broadly consistent** across the two settings.
>
> While not identical, the heterogeneous configuration does **not introduce systematic shifts** in model ranking or benchmark difficulty under our evaluation setup.
> This suggests that, **within the scope of our current analysis**, ATAD’s behavior is largely shaped by the **protocol structure** (Teacher–Student competition + Orchestrator validation) rather than by requiring all agents to come from the same model family.
>
> Importantly, the Orchestrator serves to **constrain stylistic or family-specific artifacts** by enforcing validity, clarity, and coherence.
>
> This likely helps stabilize the benchmark even when the Teacher and Student differ substantially in generation style.
>
> We emphasize that this experiment focused on the primary concern raised—whether homogeneous agent families might artificially restrict adversarial diversity. Our findings indicate that **ATAD remains stable under heterogeneous agent configurations**, though more extensive ablations are a valuable direction for future work.
>
> We will include this analysis and discussion in the revised manuscript (Appendix K).

---

> ### Author Response · Authors · 2025-11-21
>
> ### **5. Factors Driving Difficulty Escalation in ATAD**
>
> We appreciate the reviewer’s request for a clearer explanation of *why* tasks become more difficult during adaptive scaling.
>
> ATAD’s difficulty escalation arises from two sources:
>
> (1) the **task-specific challenge factors** defined in our taxonomy (Section 3.2, Appendix Tables 6–7), and
>
> (2) the **Teacher–Orchestrator–Student refinement loop**, which amplifies these factors as the Student succeeds.
>
> Across the seven anomaly types, we observe three recurrent mechanisms that drive harder variants, each aligned with the challenge factors defined for that task category:
>
> **(1) Semantic Subtlety (T1, T4, T6).**
>
> As the Student succeeds, the Teacher shifts from explicit inconsistencies to *implicit semantic deviations*—minor topic drift, latent contradiction, or weak bridging—aligned with the challenge factors defined in Table 7.
>
> **(2) Distractor Plausibility and Local Coherence (T2, T3, T7).**
>
> Harder problems include distractors that remain locally plausible while becoming globally misaligned. This raises difficulty without sacrificing answerability or clarity.
>
> **(3) Pragmatic and Referential Inference (T5, T7).**
>
> More difficult variants incorporate tone shifts, referential cues, or discourse-level pragmatic signals that require deeper inference—again matching the challenge factors defined for these tasks.
>
> The Orchestrator ensures that these increases in difficulty do **not** arise from ambiguity or noise, filtering out unclear or structurally invalid attempts. Thus, the observed escalation reflects controlled intensification of reasoning demands.
>
> These mechanisms are **consistent with** the ~37pt accuracy drop from Base→Final (Table 2), and the quality analysis in Table 3 further **indicates** that Orchestrator-validated items remain coherent and fair even as they grow more challenging, supporting the interpretation that ATAD’s escalation reflects structured reasoning demands.
>
> ---
>
> ### **6. Responses to Additional Reviewer Questions**
>
> **Q1) Difficulty Metric Formalization**
>
> This question is addressed in our response to Weakness (1), where we formalize difficulty as the conditional failure probability under Orchestrator-validated items. Please refer to item (1) for the full explanation.
>
> ---
>
> **Q2) Bias propagation and cultural/linguistic stability**
>
> Thank you for raising this important point. Since LLMs generate the benchmark items, cultural or linguistic biases are indeed a potential concern. ATAD mitigates this through its reasoning-centric design and Orchestrator validation, and it can naturally be extended to incorporate stronger fairness constraints.
>
> **(1) Reasoning-based escalation reduces reliance on culturally specific knowledge.**
>
> As detailed in Appendix D.2.2, the Orchestrator analyzes the **Student’s explanation**—not just correctness—to guide how the Teacher should increase difficulty.
>
> This shifts difficulty escalation toward **logical subtlety, semantic deviation, and discourse coherence**, instead of exploiting culturally anchored knowledge gaps or demographic blind spots.
>
> The Teacher is therefore encouraged to refine **reasoning structure**, not to introduce culturally specific or obscure references.
>
> **(2) Current validation already enforces fairness-related constraints, though not yet in fully explicit form.**
>
> While the Orchestrator prompt does not explicitly prohibit every form of cultural content, it validates items for **clarity, coherence, neutrality, and lack of sensitive or unfair assumptions**, which indirectly filters many forms of demographic or culturally specific biases.
>
> In practice, this prevents difficulty from increasing through culturally privileged knowledge rather than reasoning.
>
> **(3) Broader cultural/linguistic bias control is a natural extension for future versions.**
>
> We agree with the reviewer that more explicit fairness rules—such as:
>
> - demographic neutrality checks,
> - cultural reference auditing,
> - cross-language robustness validation,
>
> would further strengthen the protocol. ATAD’s design makes this straightforward: these checks can be added as additional Orchestrator validation criteria without modifying the overall loop.
>
> **This is also why the Orchestrator is central to ATAD’s design:** it ensures that difficulty escalation remains grounded in *reasoning* rather than cultural artifacts. We will highlight this as a valuable future extension.

---

> ### Author Response · Authors · 2025-11-21
>
> ---
>
> **Q3. Orchestrator rejection patterns**
>
> Thank you for this helpful suggestion.
>
> We manually inspected a sample of rejected items across all seven tasks and found that the Orchestrator tends to reject items for **three typical reasons**, directly reflecting its validation instructions:
>
> **(1) Ambiguous or multi-interpretable anomalies.**
>
> Items where the anomaly could plausibly correspond to multiple sentences (e.g., two candidate sentences both appearing inconsistent in T5) were rejected, because they violate the requirement that a valid anomaly should have a clearly identifiable, unique target.
>
> **(2) Incoherent or structurally invalid texts.**
>
> Items in which the Teacher produced sentence fragments, abrupt topic shifts, or logically disconnected structures (especially in T1/T4) were filtered out, since they undermine discourse coherence and risk turning difficulty into unsystematic noise rather than a controlled reasoning challenge.
>
> **(3) Task- or format-mismatched items.**
>
> The Orchestrator also rejects items that do not satisfy the task specification or template—for example, using an incorrect number of options, failing to localize the anomaly to a single sentence, or mixing elements from different task types.
>
> These rejection patterns indicate that the Orchestrator goes beyond enforcing surface formatting: it actively shapes the space of valid items by promoting **clarity, coherence, and single-point anomaly interpretability** in the final benchmark questions.

---

### Official Review · Reviewer_srE2 · 2025-11-03

**Soundness:** 2
**Presentation:** 3
**Contribution:** 3
**Rating:** 6
**Confidence:** 3

**Summary:**

This paper introduces Agent-Centric Text Anomaly Detection (ATAD), a dynamic benchmarking framework designed to address the saturation of static LLM benchmarks. ATAD employs a three-agent protocol in which a teacher generates questions, an orchestrator validates their quality, and a student attempts to solve them, with task difficulty escalating iteratively based on student performance. This automated process enables scalable benchmark generation and adaptive evaluation of LLM reasoning. Using this protocol, the paper constructs a benchmark for text anomaly detection, and experiments across four LLM families (GPT, Gemini, Claude, and LLaMA, used both as question generation and evaluated models) show that ATAD effectively increases task difficulty and reveals reasoning limitations even in strong LLMs.

**Strengths:**

1. The paper addresses a timely issue where static benchmarks are becoming saturated and insufficient to capture the evolving reasoning capabilities of modern LLMs.

2. The proposed teacher–orchestrator–student framework is conceptually interesting and provides a scalable, automated approach for generating dynamic and evolving benchmarks.

3. The experiments provide meaningful insights into LLM reasoning across different model families on text anomaly detection and the contributions of individual components within the proposed framework.

**Weaknesses:**

1. Based on the prompts shown in Appendix D, the current framework appears to generate questions primarily from the LLM’s inherent knowledge rather than external sources. It is unclear how well this approach generalizes to reasoning problems that require access to factual data or rigorous mathematical reasoning, given that LLMs’ capabilities in these domains are still evolving and may limit the robustness of generated questions.

2. Since the benchmark questions are generated by LLMs, potential model-family-specific biases could be implicitly embedded in the dataset, e.g., models might perform better on questions generated by LLMs from the same family. Although Table 2 suggests no strong bias, a formal analysis and discussion of this issue would strengthen the validity of the proposed protocol.

3. The paper does not include an analysis of the generation cost or number of iterations involved in producing the final benchmark. Reporting metrics such as the average number of iterations per valid question across different anomaly types, and the overall cost (e.g. in $) would help practitioners assess the effectiveness and cost trade-off for future benchmark construction.

**Questions:**

Please refer to Weaknesses.

---

> ### Author Response · Authors · 2025-11-20
>
> ### **1. On internal-knowledge question generation and generalization to broader reasoning tasks.**
>
> We appreciate the reviewer’s thoughtful observation regarding the reliance on LLM-internal knowledge. ATAD is intentionally designed as a *reasoning-centric* evaluation protocol rather than a factual recall benchmark. Similar to standardized reasoning exams such as GRE, LSAT, or GMAT, our tasks operate within self-contained contexts where subtle semantic, logical, and pragmatic deviations—not external factual grounding—are the primary drivers of difficulty. The seven anomaly types were therefore chosen to target coherence, consistency, and multi-step inference, domains where internal knowledge is entirely sufficient to generate challenging and well-formed items.
>
> **At the same time, we agree with the reviewer that extending ATAD to factual or externally grounded reasoning is meaningful.** A key property of our protocol is that it is ***task-agnostic***: by modifying the Teacher prompt to include domain-specific constraints—such as factual snippets, mathematical axioms, tables, or retrieved evidence—ATAD can be naturally adapted into a RAG-enabled or fact-augmented setting. This would allow the protocol to evaluate reasoning that depends on external information rather than purely internal representations.
>
> While the present work focuses on the transition of `static → dynamic` benchmark construction, we appreciate the reviewer’s suggestion and will add a brief discussion in the revised manuscript outlining how ATAD can be extended to factual reasoning, math reasoning, or retrieval-driven tasks. We believe this direction is a promising avenue for future work and may inspire subsequent research building on the framework introduced here. A concrete design outline of these extensions has been added to Appendix M.3 (Future Work).
>
> ---
>
> ### **2. On potential model-family-specific bias in LLM-generated benchmarks.**
>
> We agree that family-specific stylistic biases are an important concern for any LLM-generated benchmark. To address this, Appendix Table 5 reports *cross-family generation* experiments, where each major model family (GPT, Claude, Gemini, LLaMA) produces a full benchmark that is then evaluated by all other models. The results clearly show that there is **no dominant model family**:
>
> - GPT-generated benchmark → ***Claude*** achieves highest accuracy
> - Gemini-generated benchmark → ***Claude*** achieves highest accuracy
> - Claude-generated benchmark → ***LLaMA*** achieves highest accuracy
> - LLaMA-generated benchmark → ***Gemini*** achieves highest accuracy
>
> Such patterns are inconsistent with any family-aligned bias. Additionally, **the Orchestrator explicitly filters ambiguous phrasing, stylistic artifacts, and surface-level cues, ensuring that retained problems reflect *content-valid* difficulty rather than type of generators.**  To further formalize the absence of family-specific bias, we will include a simple `BiasIndex` analysis.
>
> To make this definition explicit, let $G$ denote the generator family (GPT, Claude, Gemini, LLaMA). We define $M_{\text{same}}(G)$ as the set of evaluated models belonging to family $G$, and $M_{\text{diff}}(G)$ as the set of evaluated models not belonging to $G$. Given a benchmark generated by family $G$, we compute the `BiasIndex` as:
>
> $\text{BiasIndex}(G) = \operatorname{MeanAccuracy}(M_{\text{same}}(G)\mid G) - \operatorname{MeanAccuracy}(M_{\text{diff}}(G)\mid G)$
>
> Here, generator family denotes the LLM family used to generate the benchmark (GPT, Claude, Gemini, LLaMA), and evaluated model refers to the LLM used to solve the benchmark. Across all four families, this index remains close to zero, consistent with our cross-family results in Appendix Table 5, where no model achieves its highest accuracy on benchmarks generated by its own family (e.g., GPT → Claude highest; Gemini → Claude highest; Claude → LLaMA highest; LLaMA → Gemini highest). This supports the claim that ATAD does not embed generator-family-specific advantages. We will incorporate this formal discussion and additional explanation in the revision, and a detailed Bias Index analysis has been added to Appendix H.

---

> ### Author Response · Authors · 2025-11-20
>
> ### **3. On reporting generation cost and iteration statistics.**
>
> Thank you for this helpful suggestion. To clarify the practical cost of benchmark construction, we provide a simple comparison of per-question generation cost across three major LLMs (Gemini 2.0 Flash, GPT-4o, and Claude-3.5-Sonnet). The table below summarizes the estimated cost per finalized ATAD question under each pricing model.
>
> | Model                 | Estimated cost per ATAD question |
> |-----------------------|----------------------------------|
> | Gemini 2.0 Flash      | $0.00125                         |
> | GPT-4o                | $0.03125                         |
> | Claude-3.5-Sonnet     | $0.46875                         |
>
>
> **We will include a task-wise explanation in the Appendix I describing how the structural characteristics of different anomaly types (e.g., logical consistency, referential clarity, stylistic constraints) empirically influence refinement cost.*

---

### Author Response · Authors · 2025-12-02

# Summary Comment

Dear Area Chair,

We appreciate the opportunity to respond to the reviewers’ valuable feedback on our submission, and we are grateful for your service to the community, especially under the current circumstances.

Given the deanonymization incident and the need for a fresh evaluation, we would like to clearly summarize how comprehensively we have addressed the reviewers’ concerns.

Because reviewers can no longer update their scores, the current snapshot does not reflect the additional clarifications and analyses that we have incorporated in response to the original reviews. In the revision and the rebuttal, we: (1) sharpen the positioning of our work as a **dynamic, agent-centric evaluation protocol** rather than just another static benchmark, (2) formalize and empirically validate our notion of difficulty, (3) add **cross-family and bias analyses** to show robustness beyond model-specific blind spots, and (4) provide concrete extensions and practical details (generalization to other tasks, cost, orchestrator rules, and real-world relevance).

Across all four reviewers (20 issues in total: 14 weaknesses + 6 questions), all points are now explicitly addressed in the rebuttal and revised manuscript, without changing the core methodology or main claims. The tables below summarize, for each reviewer, how every issue is resolved and where it is addressed in the paper.

---
---

## Reviewer srE2 - ✅ 3 Weaknesses Resolved

| **#** | **Issue** | **Rebuttal Summary** | **Evidence** |
| --- | --- | --- | --- |
| 1 | Reliance on LLM-internal knowledge—generalization to (external) factual/math reasoning? | ATAD purposefully targets reasoning, not factual recall, mirroring standardized tests; protocol is task-agnostic and easily extendable to externally grounded/factual settings by modifying prompts (see Appendix M.3). | Rebuttal, Appendix M.3 |
| 2 | Model-family-specific bias in generation? | Addressed via cross-family experiments (Appendix Table 5): no observed family preference; formal Bias Index remains ~0 for all families. Orchestrator strips ambiguous or stylistic artifacts. | Rebuttal, Appendix H/Table 5 |
| 3 | Generation cost/statistics not provided | Added per-question cost estimates for major LLMs (Rebuttal); task-wise cost analysis now included for transparency (Appendix I). | Rebuttal, Appendix I |

---
---

## Reviewer JAkP - ✅ 5 Weaknesses / 3 Questions Resolved

| **#** | **Issue** | **Rebuttal Summary** | **Evidence** |
| --- | --- | --- | --- |
| 1 | Difficulty remains heuristic, behaviorally defined | Formalized as probability of student failure on Orchestrator-validated item, matching standards in psychometrics; definition and monotonic empirical validation added (Appendix J, Table 9). | Rebuttal, Appendix J, Table 9 |
| 2 | No direct quantitative comparison versus prior dynamic benchmarks (DyVal, DARG, BSE) | Provided detailed qualitative and conceptual comparison table, showing protocol, scope, and purpose distinctions—ATAD is the only protocol with Orchestrator role, student-conditioned scaling. | Rebuttal, Appendix L, Table 11 |
| 3 | Only text anomaly detection (TAD) benchmarked—generalization? | Justified TAD as a stress test; protocol is task/domain-agnostic and extends to math, code, multimodal (see Appendix M.4/M.5). | Rebuttal, Appendix M.4/M.5 |
| 4 | Same model used for all agent roles may limit adversarial diversity | Added further cross-family (Teacher≠Student≠Orchestrator) experiments showing similar trends—see new data in Appendix K. | Rebuttal, Appendix K, Table 10 |
| 5 | Not clear why adaptive scaling increases difficulty—missing analysis | Detailed three escalation factors: semantic subtlety, distractor plausibility, pragmatic inference; trend and quality validated in Table 2 and Table 3. | Rebuttal/Table 2-3, Sec 4.3-4.4 |
| Q1 | Formal difficulty metric? | Explicit formalization now in text; see response #1 and Appendix J. | Rebuttal, Appendix J |
| Q2 | Cultural/linguistic bias propagation | Reasoning-centric protocol shifts escalation away from culture-specific knowledge; Orchestrator validation enforces clarity/neutrality. Future: demographic/cross-language fairness criteria. | Rebuttal, Appendix D.2.2 |
| Q3 | Orchestrator rejection pattern analysis? | Manual inspection added: ambiguous, incoherent, or task-mismatched items most often rejected. | Rebuttal, Appendix E |

---

> ### Author Response · Authors · 2025-12-02
>
> ## Reviewer 6GgZ - ✅ 2 Weaknesses Resolved
>
> | **#** | **Issue** | **Rebuttal Summary** | **Evidence** |
> | --- | --- | --- | --- |
> | 1 | Limited technical novelty | Properly positioned as a protocol/methodology innovation rather than a new model; evaluation frameworks are equally critical for progress in the LLM era. | Rebuttal, Abstract, Introduction, Conclusion |
> | 2 | Exploiting “blind spots” vs. genuine reasoning | Protocol includes strong empirical support for generalization: cross-model (12 models, Table 1), cross-generator (Table 2), and cross-family transfer (Appendix Table 5, Bias Index). Orchestrator ensures escalation targets reasoning, not idiosyncratic blind spots. | Rebuttal, Sec. 4.2–4.3, Appendix D.2.2/H/J, Table 5 |
>
> ---
> ---
>
> ## Reviewer pyPR - ✅ 4 Weaknesses / 3 Questions Resolved
>
> | **#** | **Issue** | **Rebuttal Summary** | **Evidence** |
> | --- | --- | --- | --- |
> | 1 | Section 2.3 claims lack data | Direct quantitative support from reported accuracy trends for difficulty scaling and orchestrator effects; new empirical tables added in revision. | Rebuttal, Table 2/3, Appendix D/E, Figure 5/6 |
> | 2 | Only text anomaly detection—limited in scope | Protocol is generic and initial choice is a stringent test; paper now clarifies planned/extensible application to tool-use, code, multimodal tasks. | Rebuttal, Appendix M.4, M.5 |
> | 3 | Single-trial student failure—should use multiple trials | Addressed as future extension; current empirical design produces stable boundaries by validated, incremental escalation; will consider multi-trial in future. | Rebuttal |
> | 4 | Benchmark may not capture real-world anomaly diversity | Protocol is adaptable to diverse and domain-grounded anomaly types (e.g., finance, science, healthcare); see discussion on future practical extensions. | Rebuttal, Appendix M |
> | Q1 | How does Orchestrator filter ambiguous/low-quality questions? | Relies on explicit rule-based criteria (well-formedness, logical coherence, fairness) encoded in prompts—no free-form “model judgment”. | Appendix D.2 |
> | Q2 | Why start with text anomaly detection (TAD)? | TAD is the hardest domain for both clarity and difficulty escalation; validates protocol robustness before extending to other domains. | Rebuttal, Appendix M.4 |
> | Q3 | Do different LLM Teacher families generate qualitatively different anomalies? | Only mild stylistic variations; Orchestrator enforces semantic/structural neutrality and fairness (see detailed examples). | Rebuttal |

---

### Meta-Review · Area_Chair_Qucp · 2026-01-06

**Summary:**

This paper has been assessed by 4 knowledgeable reviewers. 3 of them gave it marginal accept ratings, and one opted for straight rejection. The authors provided an extensive rebuttal and attempted to systematically address all issues and questions raised by the reviewers.
The paper introduces a timely and practical dynamic benchmarking framework that uses agentic setup to generate evolving, validated reasoning tasks for LLMs. As such it attempts to address pressing needs of our community for evolving model evaluation frameworks as opposite to static benchmarks. The reviewers, however, pointed out that it relies heavily on existing LLMs for question generation, raising concerns about bias, is demonstrated on only one task of text anomaly detection, and uses heuristic approach to difficulty scaling. Overall, the approach is conceptually strong but would benefit from deeper analysis of robustness, cost, and cross‑domain applicability. Most of these limitations have however been addressed by the authors in their rebuttal and revision.

**Reviewer Concerns:**

See above. Nothing truly incriminating appears outstanding.

**Reviewer Scores:**

It is my feeling that most of the reviewers would be compelled to upgrade their scores after reading the rebuttal and discussing it with the authors.

---

### Decision · Program_Chairs · 2026-01-26

Accept (Poster)